# TreeGen: A Bayesian Generative Model for Hierarchies

**Marcel Kollovieh**[1,2,3]    **Nils Fleischmann**[4]    **Filippo Guerranti**[1,2,3]

**Bertrand Charpentier**[4]    **Stephan Günnemann**[1,2,3,4]

[1] School of Computation, Information and Technology, Technical University of Munich
[2] Munich Data Science Institute    [3] Munich Center for Machine Learning    [4] Pruna AI

Correspondence to: `m.kollovieh@tum.de`

## Abstract

In this work, we introduce TreeGen, a novel generative framework modeling distributions over hierarchies. We extend Bayesian Flow Networks (BFNs) to enable transitions between probabilistic and discrete hierarchies parametrized via categorical distributions. Our proposed scheduler provides smooth and consistent entropy decay across varying numbers of categories. We empirically evaluate TreeGen on the jet-clustering task in high-energy physics, demonstrating that it consistently generates valid trees that adhere to physical constraints and closely align with ground-truth log-likelihoods. Finally, by comparing TreeGen's samples to the exact posterior distribution and performing likelihood maximization via rejection sampling, we demonstrate that TreeGen outperforms various baselines.

## 1  Introduction

Hierarchies are central to representing complex relationships across diverse domains. Consequently, hierarchical clustering algorithms are core components in well-established machine libraries such as scikit-learn [36], and find applications in a wide range of fields, spanning from phylogenetics and particle physics to web and citation network analysis. In phylogenetics, for example, clustering algorithms group organisms or genetic sequences based on similarity, helping to infer evolutionary relationships [18]. Further, real-world systems, from neural representation [22] to citation networks and web graphs [37] often exhibit hierarchical organization. Importantly, in high-energy physics, agglomerative linkage algorithms are indispensable for jet clustering and provide insights into the substructure of particle collisions [7].

Traditional hierarchical clustering algorithms operate primarily in an *unsupervised* setting. *Agglomerative* methods greedily merge clusters with the shortest distance until only a single root remains [21]. *Divisive* algorithms work in a top-down approach and start with all points in a single cluster and iteratively split them into smaller groups [43, 14]. More recent methods optimize global cost functions via continuous relaxations, yet the setting remains unsupervised [11, 34, 9, 48, 30]. This is sensible in many applications, as ground truth hierarchies are often unavailable for supervised training.

Jet clustering in high-energy physics is a prime example where supervised simulations are available, yet traditional algorithms remain unsupervised. In particle accelerators, high-energy collisions produce unstable particles that successively decay and split into more particles. This resulting spray of particles, also known as a *jet*, is detected by particle detectors. The task in jet clustering is to reconstruct the latent hierarchy that describes the splitting process from the observed constituents, i.e., leaves of the hierarchy [33]. Simulators are extensively used in this field to generate collision events based on a physical model, enabling the creation of realistic hierarchy datasets for jet clustering [3, 4,

12]. Yet, despite this availability, prevailing jet-clustering pipelines often still rely on unsupervised algorithms [15, 46, 7, 17].

To address this problem, we introduce a novel generative model for hierarchies. Building on Bayesian Flow Networks (BFNs) [19] and Bayesian Sample Inference (BSI) [31], our model explicitly models the posterior distribution over hierarchies.

Our *key contributions* are summarized as follows:

- **Generative Model for Hierarchies.** We introduce TreeGen, a novel generative model tailored to tree-structured data. The model transitions between probabilistic and discrete hierarchies parametrized via categorical distributions.
- **Entropy Scheduler for BFNs.** We propose an alternative entropy scheduler for BFNs that provides smooth transitions of categorical distributions from uniform to discrete states across varying numbers of categories.
- **Application to Jet Clustering.** We demonstrate the practical effectiveness of our generative model by applying it to the jet clustering task using data from the GINKGO simulator [12].

## 2 Background

### 2.1 Probabilistic Hierarchies

Let $\hat{\mathcal{T}}$ be a rooted hierarchy, also called a tree, with $n$ leaves $V = \{v_1, \ldots, v_n\}$ and $n'$ internal nodes $Z = \{z_1, \ldots, z_{n'}\}$, where $z_{n'}$ is the root. We represent the hierarchy via two binary adjacency matrices:

$$\hat{\boldsymbol{A}} \in \{0,1\}^{n \times n'}, \quad \hat{\boldsymbol{B}} \in \{0,1\}^{n' \times n'}, \quad (1)$$

where the entry $\hat{\boldsymbol{A}}_{ij}$ encodes an edge from leaf $v_i$ to internal node $z_j$, and $\hat{\boldsymbol{B}}_{ij}$ encodes an edge from internal node $z_i$ to internal node $z_j$.

We enforce both matrices to be row-stochastic and additionally constrain $\hat{\boldsymbol{B}}$ to be upper triangular to ensure they define a valid tree-structure. In the binary case, this means that each row contains exactly one non-zero entry, while the last row of $\hat{\boldsymbol{B}}$ is all zeros

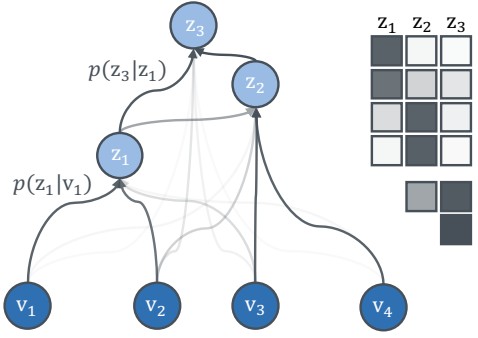

Figure 1: Example of a probabilistic hierarchy. The matrices represent $\boldsymbol{A}$ and $\boldsymbol{B}$, respectively.

because the root has no parent. These constraints guarantee that $(\hat{\boldsymbol{A}}, \hat{\boldsymbol{B}})$ encodes a valid rooted tree.

Following Zügner et al. [48], we relax these matrices to *probabilistic* assignments:

$$\boldsymbol{A} \in [0,1]^{n \times n'}, \quad \boldsymbol{B} \in [0,1]^{n' \times n'}, \quad (2)$$

while retaining the row-stochastic and upper-triangular constraints. We interpret each row of $\boldsymbol{A}$ and $\boldsymbol{B}$ as a categorical distribution over parent assignments:

$$\boldsymbol{A}_{ij} = p(z_j \mid v_i), \quad \boldsymbol{B}_{ij} = p(z_j \mid z_i), \quad (3)$$

denoting the probability that internal node $z_j$ is the parent of leaf $v_i$ and the probability that $z_j$ is the parent of internal node $z_i$, respectively. Together, the matrices describe a *probabilistic hierarchy* $\mathcal{T} = (\boldsymbol{A}, \boldsymbol{B})$. By interpreting each row of a probabilistic hierarchy as a categorical distribution, we can sample valid discrete hierarchies. An example of a probabilistic hierarchy is shown in Fig. 1.

### 2.2 Bayesian Flow Networks

Bayesian Flow Networks (BFNs) form a class of generative models that refine a belief over a target sample via successive Bayesian updates [19]. Recently, Lienen et al. [31] proposed Bayesian Sample

Project page: cs.cit.tum.de/daml/treegen

Inference (BSI), including BFNs in a more general framework, which we adopt here for notational simplicity. BSI starts with an initial belief over $x \in \mathbb{R}^D$ using a Gaussian prior with precision $\lambda$,

$$p(x) = \mathcal{N}(\mu, \lambda^{-1}), \quad \mu \in \mathbb{R}^D, \ \lambda > 0, \tag{4}$$

which is updated in a Bayesian manner: $p(x \mid y) \propto p(y \mid x)p(x)$. At each step, we observe a noisy measurement

$$y \sim \mathcal{N}(x, \alpha^{-1}), \quad \alpha > 0, \tag{5}$$

where $\alpha$ denotes the precision of the measurement. We compute the closed-form Gaussian posterior

$$p(x \mid y) = \mathcal{N}\left(\tfrac{\lambda\mu + \alpha y}{\lambda + \alpha}, (\lambda + \alpha)^{-1}\right), \tag{6}$$

which converges towards the true $x$ as the precision $\lambda$ increases by $\alpha$ with each update. As we do not have direct access to the sample noisy measurements $y$ of $x$, we train a neural network $\hat{x} = f_\theta(\mu, \lambda)$ to approximate $x$. The model is trained similarly to diffusion models [23] and minimizes the squared error between $\hat{x}$ and the ground truth $x$. However, unlike diffusion models, BFNs and BSI operate on distributions rather than single samples. While BSI has considered the Gaussian case, it is not limited to it, and we will consider categorical distributions in Sec. 3, as done by BFNs.

## 3 TreeGen: A Generative Model for Hierarchies

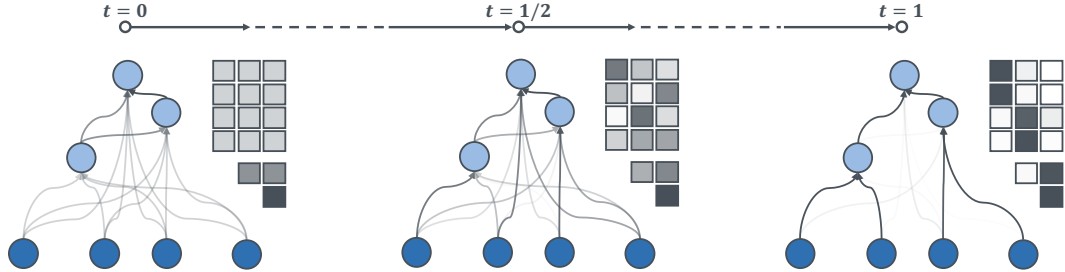

Figure 2: Overview of the TreeGen generation process. At each continuous time $t \in [0, 1]$, the model maintains a probabilistic hierarchy $\mathcal{T}_t$, applies a Bayesian-inspired update via a neural network and noisy sample, and progressively reduces entropy until converging to a discrete hierarchy $\hat{\mathcal{T}}$ at $t = 1$.

In this section, we present our main contribution: TreeGen, a Bayesian generative model designed for hierarchies. TreeGen starts from a probabilistic hierarchy $\mathcal{T}$, which is iteratively updated via Bayesian updates of observed noisy hierarchies until the entropy of the hierarchy is sufficiently low and converges toward a discrete hierarchy. To obtain noisy observations, we design a neural network trained to predict ground-truth discrete hierarchies. As such a hierarchy simply consists of a collection of categorical variables (see Sec. 2.1), we first describe the framework for arbitrary categorical distributions. While we derive update equations within the BSI framework [31], the resulting Bayesian updates match those of BFNs [19] but follow from a simpler derivation. Compared with standard BFNs, TreeGen instantiates a categorical BSI that uses the BFN update with explicit precision control while introducing a different loss, omits auxiliary distributions, and extends the parametrization to trees. We discuss these differences and provide detailed derivations in App. A.

**Objective and prior.** Our goal is to infer an unknown sample $x \in \{1, \ldots, K\}$. We encode our current belief about that sample via a categorical prior, i.e.,

$$p(x = k) = \pi_k, \qquad \boldsymbol{\pi} = (\pi_1, \ldots, \pi_K) \in \Delta_{K-1}, \tag{7}$$

where $\Delta_{K-1}$ is the $K$-simplex, i.e., $\pi_k \geq 0$ and $\sum_k \pi_k = 1$.

**Noisy observation.** Assume we have access to a noisy measurement vector $\mathbf{y} = (y_1, \ldots, y_K)$ sampled from a multinomial distribution with $m = \sum_k y_k$ trials and class probabilities $\boldsymbol{\theta}$, which we model as a mixture between the ground-truth Dirac distribution $\mathbf{e}_x$ and uniform noise:

$$\boldsymbol{\theta} = \omega \, \mathbf{e}_x + (1 - \omega) \tfrac{1}{K} \mathbf{1}, \qquad 0 \leq \omega \leq 1. \tag{8}$$

$\mathbf{e}_x$ is the one-hot encoding of $x$ and $\omega$ defines a signal–noise trade-off: $\omega = 1$ yields perfectly informative counts, while $\omega = 0$ gives pure uniform noise.

The conditional likelihood for $x = k$ is

$$p(\mathbf{y} \mid x = k, \omega) = \frac{m!}{\prod_j y_j!} \left(\tfrac{1-\omega}{K}\right)^{m-y_k} \left(\tfrac{1-\omega}{K} + \omega\right)^{y_k}. \tag{9}$$

**Bayes update.** After observing $\mathbf{y}$, we can update our belief on $x$, i.e., parameters $\boldsymbol{\pi}$ via Bayes' rule:

$$p(x = k \mid \mathbf{y}, \omega) = \frac{p(\mathbf{y} \mid x = k, \omega)\,\pi_k}{\sum_j p(\mathbf{y} \mid x = j, \omega)\,\pi_j}. \tag{10}$$

By canceling the common multinomial coefficients, we obtain a softmax-type expression:

$$p(x = k \mid \mathbf{y}, \omega) = \frac{\pi'^{y_k}\,\pi_k}{\sum_j \pi'^{y_j}\,\pi_j}, \qquad \pi' := 1 + \frac{\omega K}{1-\omega}. \tag{11}$$

We can express this equivalently in vectorized form:

$$\boldsymbol{\pi}_{\text{post}} = \text{softmax}\big(\log \boldsymbol{\pi} + \mathbf{y} \log \pi'\big). \tag{12}$$

This form shows that the observation contributes an additive vector $\mathbf{y} \log \pi'$ to our belief before the softmax normalization. This form also allows simple aggregation of sequential updates.

**Sequential updates.** With multiple independent observations $\{\mathbf{y}^{(i)}\}_{i=1}^N$ and constant $\omega$, the posterior update simplifies to:

$$\boldsymbol{\pi}^{(N)} = \text{softmax}\Big(\log \boldsymbol{\pi}^{(0)} + \log \pi' \sum_{i=1}^N \mathbf{y}^{(i)}\Big). \tag{13}$$

Thus, the class probabilities are aggregated by addition in log-space.

**Gaussian interpretation.** For a large observation count $m$, the Central Limit Theorem implies that the rescaled sum of counts is approximately Gaussian. This allows us to rewrite the Bayesian update using a Gaussian-distributed variable and in continuous time:

$$\boxed{\boldsymbol{\pi}_t = \text{softmax}\big(\log \boldsymbol{\pi}_{t-1} + \mathbf{z}_t\big), \qquad \mathbf{z}_t \sim \mathcal{N}\big((\alpha_t - \alpha_{t-1})K\mathbf{e}_x, (\alpha_t - \alpha_{t-1})K\mathbf{I}\big).} \tag{14}$$

Due to the additive aggregation in log-space (see Eq. (13)), we can summarize multiple Bayesian updates into a single one, i.e., the step from $\boldsymbol{\pi}_0$ to $\boldsymbol{\pi}_t$:

$$\boxed{\boldsymbol{\pi}_t = \text{softmax}\big(\mathbf{z}_t\big), \qquad \mathbf{z}_t \sim \mathcal{N}(\alpha_t K \mathbf{e}_x, \alpha_t K \mathbf{I}),} \tag{15}$$

where $\alpha_t$ represents the accumulated signal. Note that we omitted $\log \boldsymbol{\pi}_0$ as it is constant and the softmax operation is shift invariant. Consequently, the log-evidence in the softmax follows a Gaussian and is parametrized with $\alpha_t$. The recovered updates equal those of Graves et al. [19].

**Distribution of $\boldsymbol{\pi}_t$.** Following the categorical BFN formulation, we represent the belief vector at time $t$ as the softmax of latent Gaussian logits, $\boldsymbol{\pi}_t = \text{softmax}(\mathbf{z}_t)$ conditioned on the ground-truth class $x$. This results in the distribution:

$$p(\boldsymbol{\pi}_t \mid \boldsymbol{\pi}_{t-1}, \mathbf{x}) = \int \delta\big(\boldsymbol{\pi}_t - \text{softmax}(\mathbf{z}_t)\big)\,p(\mathbf{z}_t \mid \mathbf{x}, \alpha_t, \alpha_{t-1})\,d\mathbf{z}_t, \tag{16}$$

Furthermore, we can model the marginal of $\boldsymbol{\pi}_t$ directly without intermediate steps:

$$p(\boldsymbol{\pi}_t \mid \mathbf{x}) = \int \delta\big(\boldsymbol{\pi}_t - \text{softmax}(\mathbf{z}_t)\big)\,p(\mathbf{z}_t \mid \mathbf{x}, \alpha_t)\,d\mathbf{z}_t. \tag{17}$$

Both distributions are trivial to sample by drawing $\mathbf{z}_t$ from the corresponding Gaussian and applying the softmax. During generation, the true class $x$ is unknown, so we replace it with a proxy $\hat{x} = f_{\boldsymbol{\theta}}(\boldsymbol{\pi}_t, t)$ predicted by our neural network and approximate the one-step transition by $p_{\boldsymbol{\theta}}(\boldsymbol{\pi}_t \mid \boldsymbol{\pi}_{t-1}) = p(\boldsymbol{\pi}_t \mid f_{\boldsymbol{\theta}}(\boldsymbol{\pi}_t, t))$, which recovers the distribution of $\boldsymbol{\pi}_t$ when $f_{\boldsymbol{\theta}}(\boldsymbol{\pi}_t, t)$ successfully predicts $x$. Unlike the BFN update, this allows us to control the precision directly.

**Training of the neural network $f_\theta(\boldsymbol{\pi}_t, t)$.** Our goal is to train the network to recover the original sample $x$ from a categorical distribution $\boldsymbol{\pi}_t$. Therefore, at each training step we draw a ground-truth sample $x$ from the dataset $\mathcal{D}$ and a timestep $t \sim \mathcal{U}(0, 1)$, then sample the corresponding distribution $\boldsymbol{\pi}_t$ according to Eq.(15). The network receives $\boldsymbol{\pi}_t$ and $t$ as inputs, and outputs a categorical distribution, aiming to concentrate all probability mass on the true class $x$. We optimize the parameters $\theta$ by minimizing the expected cross-entropy loss:

$$\mathcal{L}(\theta) = \mathbb{E}_{x \sim \mathcal{D}, t \sim \mathcal{U}(0,1), \boldsymbol{\pi}_t \sim p_t(\boldsymbol{\pi}_t | \mathbf{x})} \Big[ \text{CELoss}\big(f_\theta(\boldsymbol{\pi}_t, t), x\big) \Big], \quad (18)$$

where CELoss denotes the cross-entropy loss between the predicted categories and the ground-truth label. The nested expectation is approximated with Monte-Carlo samples of $x$, $t$, and $\boldsymbol{\pi}_t$ in each minibatch, and gradients are propagated through the network to update $\theta$.

**Entropy scheduler.** During generation, we want the entropy of $\boldsymbol{\pi}_t$ to decrease smoothly from its maximum (uniform) value to zero (one-hot). If we draw

$$\mathbf{z}_t \sim \mathcal{N}\big(\alpha_t K \mathbf{e}_x, \alpha_t K \mathbf{I}\big),$$
$$\boldsymbol{\pi}_t = \text{softmax}(\mathbf{z}_t),$$

then the entropy of $\boldsymbol{\pi}_t$ is a monotone function of $\alpha_t$. While BFNs [19] chose $\alpha_t = Ct^2$, the constant $C$ requires retuning when the number of classes $K$ changes. We instead propose

$$\alpha_t = -\frac{C + a \log_2 K}{K} \ln(1 - t), \quad (19)$$

for $t \in [0, 1]$, which yields an *approximately linear* decay of expected normalized

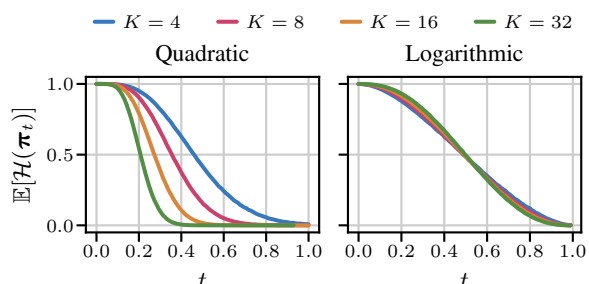

Figure 3: Comparison of entropy schedulers. The plots show the expected normalized entropy of the categorical belief $\boldsymbol{\pi}_t$ over time $t$, for varying class counts $K$. **Left:** the quadratic schedule used in standard BFNs. **Right:** our proposed log-based schedule, which yields an approximately linear decay across different $K$.

entropy (see Fig. 3) while ensuring $\boldsymbol{\pi}_0$ is uniform and $\boldsymbol{\pi}_1 = \mathbf{e}_x$. Because our experiments span hierarchies with varying numbers of classes (see Sec. 2.1), this scheduler enables us to use the *same* hyperparameters across all sizes of hierarchies.

**Adaptation to hierarchies.** Extending the categorical generation from a single random variable to an entire hierarchy means treating every row of $(\boldsymbol{A}, \boldsymbol{B})$ as its own categorical distribution, while updating all rows *jointly* so that the resulting sample is a valid hierarchy. Concretely, we model the $n$ rows of $\boldsymbol{A}$ and the $n'$ rows of the upper-triangular matrix $\boldsymbol{B}$ as categorical distributions, where each class represents a parent choice for a node. Note that $K$, i.e., the number of possible parents, varies with hierarchy size and across rows of $\boldsymbol{B}$ due to the upper-triangular structure. This parametrization induces a distribution over rooted trees (hierarchies) and substantially reduces the search space compared to arbitrary graph generation. Accordingly, $f_\theta$ takes the current probabilistic hierarchy $\mathcal{T}_t = (\boldsymbol{A}_t, \boldsymbol{B}_t)$ and outputs a discrete proposal $\hat{\mathcal{T}}$. This prediction is then used to update the current belief, i.e., the probabilistic hierarchy, using Eq. (15). We show algorithms for sampling and training in Algs. 1 and 2, respectively.

| **Algorithm 1** Sampling with TreeGen | **Algorithm 2** Training TreeGen |
|---|---|
| **input:** Neural network $f_\theta$, entropy schedule $\alpha_t$ for $t \in [0, 1]$, sampling steps $N$ | **input:** Dataset $\mathcal{D}$, neural network $f_\theta$, entropy schedule $\alpha_t$ for $t \in [0, 1]$, training steps $N$ |
| **output:** Discrete hierarchy $\mathcal{T}_1$ | 1: **for** $n = 1$ **to** $N$ **do** |
| 1: $\mathcal{T}_0 = 1/K$ | 2: $\quad \mathcal{T} \sim \mathcal{D}, t \sim \mathcal{U}(0, 1)$ |
| 2: **for** $t$ **in** $\{1/N, \dots, 1\}$ **do** | 3: $\quad \mathbf{z}_t \sim \mathcal{N}\big(\alpha_t K \mathcal{T}, \alpha_t K \mathbf{I}\big) \quad \triangleright$ Eq. (15) |
| 3: $\quad \hat{\mathcal{T}} \leftarrow f_\theta(\mathcal{T}_{t-1/N}, t - 1/N) \quad \triangleright$ Evaluate $f_\theta$ | 4: $\quad \mathcal{T}_t \leftarrow \text{softmax}(\mathbf{z}_t) \quad \triangleright$ Eq. (15) |
| 4: $\quad \mathbf{z}_t \sim \mathcal{N}\big(\alpha_t K \hat{\mathcal{T}}, \alpha_t K \mathbf{I}\big) \quad \triangleright$ Eq. (15) | 5: $\quad \hat{\mathcal{T}} \leftarrow f_\theta(\mathcal{T}_t, t) \quad \triangleright$ Evaluate $f_\theta$ |
| 5: $\quad \mathcal{T}_t \leftarrow \text{softmax}(\mathbf{z}_t) \quad \triangleright$ Eq. (15) | 6: $\quad \mathcal{L} \leftarrow \text{CE}(\hat{\mathcal{T}}, \mathcal{T}) \quad \triangleright$ Compute loss |
| 6: **end for** | 7: $\quad \theta \leftarrow \theta - \eta \nabla_\theta \mathcal{L} \quad \triangleright$ Gradient step |
| 7: **return** $\mathcal{T}_1$ | 8: **end for** |

# 4 Experiments

In this section, we present our empirical results on the jet-clustering task. Our primary goal is to demonstrate the effectiveness of TreeGen in generating valid hierarchies adhering to the true data distribution. We compare TreeGen against various baselines and test how well the generated hierarchies approximate the ground-truth posterior distributions.

We evaluate TreeGen on hierarchies derived from the GINKGO dataset, focusing on QCD and W jets. More specifically, our experiments focus on the conditional generation $p(\mathcal{T} \mid \boldsymbol{X})$, where $\boldsymbol{X}$ are leaf features. Finally, we conduct ablation studies to analyze the impact of key design choices in our method in App. C.

## 4.1 Experimental Setup

**Datasets.** Our evaluation uses five datasets obtained with the GINKGO jet shower generator [12]: **QCD jets:** QCD-S, QCD-M, QCD-L, and **W-Boson jets:** W-S, W-M. All five share the same root four-momentum $p_{\text{root}}^\mu$ and decay rate $\lambda$, but vary in the shower cut-off $\Delta_{\text{cut}}$, controlling the final jet size. The datasets have $\Delta_{\text{cut}} \in \{4.0^2, 1.1^2, 0.6^2\}$ and contain hierarchies with at most 19 (small), 59 (medium), and 99 (large) nodes. Each dataset consists of 100,000 hierarchies. Split into 98,000 hierarchies for training and 1,000 for validation and testing. In our setup, the leaves of the hierarchies correspond to the observed particles with features $\boldsymbol{X}$, while internal nodes are latent and to be inferred via the sampled hierarchy $\mathcal{T}$. Fig. 8 shows the leaf-count distribution for the three QCD datasets. We provide more details about the datasets and task in App. B.2.

**Baselines.** We benchmark the generative capabilities of TreeGen against various baselines. We include CatFlow [16] and (standard) Bayesian Flow Networks (BFNs) [19] representing state-of-the-art generative models for categorical data. Furthermore, we compare to greedy clustering approaches and the ground-truth posterior obtained via cluster trellis [33]. Finally, we compare to the CA [15, 46], $k_T$ [17], and anti-$k_T$ [7] algorithms in App. C, representing traditional jet clustering algorithms. All generative baselines share the tree assumption (see Sec. 2.1), while the agglomerative algorithms are tailored to jet clustering and restricted to binary hierarchies. We describe the baselines in more detail in App. B.5.

**Evaluation metrics.** To assess the performance of the generative model, we employ two metrics. First, we compute the — valid hierarchies — fraction of generated hierarchies that satisfy dataset-specific constraints, i.e., required physical properties. Second, we assess how well the sampled hierarchies match the data distribution by comparing their log-likelihoods to those of the ground-truth hierarchies. Specifically, we compute the ratio of log-likelihoods of each generated hierarchy to its corresponding ground-truth counterpart and report the average ratio over the test set. A ratio close to one indicates that the model closely approximates the true data distribution. We discuss further details about both metrics in App. B.6.

**Practical considerations.** TreeGen requires an architecture that predicts a hierarchy given a probabilistic hierarchy as input. This structure makes it suitable for a Graph Transformer [38]. We fix $n' = n - 1$, i.e., one parent for every non-root node, as we operate on binary hierarchies. We depict the high-level architecture in Fig. 4 and provide more details in App. B.3 and discuss node and edge features in App. B.4. We train all models using Adam [28] with a learning rate of 0.0001 and gradient clipping set to 0.5 for 50 epochs. For generation, all models use 1000 steps, except for the ablation in Fig. 7, where all use 100. We provide an overview of all hyperparameters in App. B.1. Finally, we report the mean and standard deviation of four random seeds for all experiments to ensure reproducibility.

## 4.2 Results

**Hierarchy generation.** We report the fraction of valid trees and log-likelihoods in Tab. 1. TreeGen generates *nearly perfect* trees in every case, achieving $\geq 0.93$ validity and likelihood ratios close to one, consistently outperforming baselines across all datasets. By contrast, CatFlow and BFN deteriorate rapidly as jet size grows: on the medium QCD set, their valid-tree fractions drop to 0.24 and 0.65, respectively. We attribute CatFlow's inferior performance to its flow leaving the

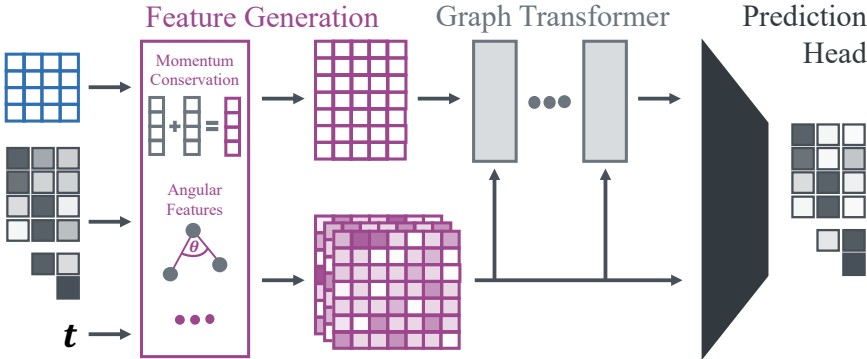

Figure 4: TreeGen model architecture. Node and edge features (structural and physics-inspired) are derived from the current probabilistic hierarchy, then processed by a Graph Transformer to predict the parameters of the final hierarchy $\mathcal{T}_1$.

probability simplex: intermediate states can violate the categorical constraints that define a valid jet hierarchy. The Bayesian-based approaches BFN and TreeGen remain on the simplex throughout generation by design. These results show that TreeGen's successfully scales to larger, more complex jets, consistently maintaining both the highest validity and the best log-likelihood fractions.

Table 1: Evaluation metrics for different models across datasets. Best scores in **bold**.

| Dataset | CatFlow | | BFN | | TreeGen | |
|---|---|---|---|---|---|---|
| | Valid Frac. (↑) | LLH Frac. | Valid Frac. (↑) | LLH Frac. | Valid Frac. (↑) | LLH Frac. |
| QCD-S | $0.752_{\pm0.014}$ | $0.972_{\pm0.002}$ | $0.941_{\pm0.003}$ | $0.982_{\pm0.000}$ | $\mathbf{0.997}_{\pm0.001}$ | $\mathbf{1.003}_{\pm0.002}$ |
| QCD-M | $0.236_{\pm0.016}$ | $0.882_{\pm0.003}$ | $0.645_{\pm0.040}$ | $0.936_{\pm0.002}$ | $\mathbf{0.977}_{\pm0.010}$ | $\mathbf{0.994}_{\pm0.002}$ |
| QCD-L | - | - | $0.416_{\pm0.036}$ | $0.898_{\pm0.002}$ | $\mathbf{0.943}_{\pm0.016}$ | $\mathbf{0.975}_{\pm0.001}$ |
| W-S | $0.604_{\pm0.014}$ | $0.937_{\pm0.007}$ | $0.851_{\pm0.014}$ | $0.965_{\pm0.001}$ | $\mathbf{0.994}_{\pm0.002}$ | $\mathbf{1.006}_{\pm0.001}$ |
| W-M | - | - | $0.341_{\pm0.054}$ | $0.886_{\pm0.002}$ | $\mathbf{0.930}_{\pm0.024}$ | $\mathbf{0.980}_{\pm0.004}$ |

Furthermore, Fig. 5 visualizes the *per-tree* log-likelihoods: each dot compares the log-likelihood of the ground-truth hierarchy (x-axis) with the log-likelihood of the hierarchy generated by our model (y-axis). TreeGen's points cluster along the diagonal, indicating alignment with the true distributions. This confirms that TreeGen not only produces more valid hierarchies but faithfully reproduces the likelihoods of the ground-truth.

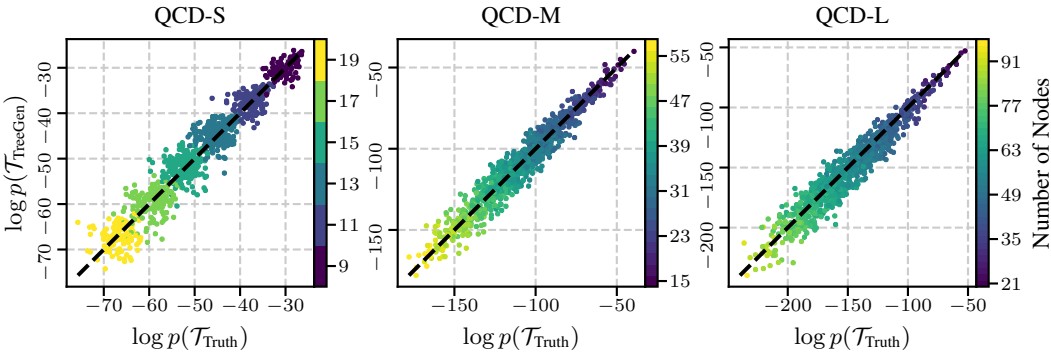

Figure 5: Scatter plot of log-likelihoods. Each point corresponds to a test hierarchy: the $x$-axis shows the log-likelihood of true hierarchy, and the $y$-axis shows the log-likelihood hierarchy generated by TreeGen, demonstrating close alignment across varying hierarchy sizes.

**Posterior distribution.** While our reported quantitative metrics demonstrate that TreeGen produces reasonable hierarchies, they do not directly assess how well it approximates the true posterior $p(\mathcal{T} \mid X)$. To complement our evaluation, we employ the cluster trellis data structure [33] to draw exact samples from the posterior. This enables us to compare these samples with those generated by TreeGen. Since the computational complexity of cluster trellis scales exponentially as $\mathcal{O}(3^N)$, this evaluation is only feasible for the QCD-S dataset, which contains hierarchies of up to 20 nodes. For each of the 1000 test hierarchies, we draw one sample from the true posterior via cluster trellis and one from TreeGen.

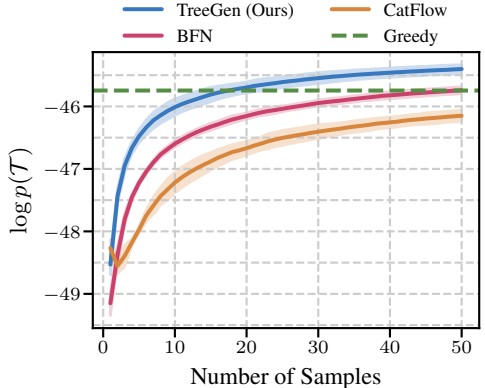

Figure 6: Histogram of posterior log-likelihoods. Distributions of $\log p(\mathcal{T} \mid X)$ for true hierarchies versus TreeGen-sampled hierarchies, demonstrating that TreeGen successfully approximates the full posterior.

We visualize and compare the empirical distributions over the log-likelihoods using histograms in Fig. 6. The empirical distribution produced by our generative model successfully captures the key characteristics of the true posterior distribution. This further validates TreeGen's ability to accurately learn and represent hierarchical structures.

**Likelihood Maximization.** While sampling from TreeGen provides diverse plausible reconstructions, many applications require a single high-likelihood hierarchy. Our model's non-deterministic nature allows us to resample hierarchies multiple times. To this end, we perform approximate MAP inference via rejection sampling. More specifically, for each jet event $X$, we draw $N$ independent hierarchies $\{\hat{\mathcal{T}}_i\}_{i=1}^N \sim p_\theta(\hat{\mathcal{T}} \mid \mathbf{X})$, compute their log-likelihoods $\log p(\hat{\mathcal{T}}_i)$, and select the sample with the highest value. We benchmark this against a deterministic greedy agglomerative linkage algorithm that merges clusters by maximizing local likelihood gains. In Fig. 7, we show how the log-likelihood grows with the number of samples $N$ compared to the generative baselines and the greedy algorithm.

Figure 7: Likelihood maximization. Maximum log-likelihood obtained by drawing $N$ candidate hierarchies from TreeGen and the log-likelihood of the deterministic greedy reconstruction. TreeGen outperforms the baseline within a few samples.

We observe that the log-likelihood obtained by TreeGen exceeds that of the greedy agglomerative baseline after only 16 samples and consistently outperforms CatFlow and BFN, demonstrating that our samples not only concentrate around higher-probability hierarchies but also capture diversity beyond a single greedy pass. Note that both this rejection-sampling MAP procedure and the greedy likelihood clustering assume access to the exact likelihood, which is often unknown or intractable for realistic collision data. Nevertheless, this experiment highlights the practical benefit of our model's ability to sample multiple hierarchies, enabling improved reconstruction quality whenever quantitative evaluations are available.

**Scheduler comparison.** We compare our scheduler, proposed in Sec. 3, with the BFN scheduler [19] across different $C$. Following Graves et al. [19], we include $C = 0.75$ and $C = 3.0$. Additionally, we test $C = 6.0$ and $C = 9.0$. The results on QCD-S are shown in Tab. 2.

As we observe, the scheduler of Graves et al. [19] is highly sensitive to $C$. While $C = 0.75$ highly degrades validity, larger values result in matching performance to ours.

Table 2: Scheduler ablation on validity and LLH fraction.

| Scheduler | Valid Frac. (↑) | LLH Frac. |
|---|---|---|
| Ours | $0.997 \pm 0.001$ | $1.003 \pm 0.002$ |
| C = 0.75 | $0.410 \pm 0.019$ | $0.990 \pm 0.002$ |
| C = 3.0 | $0.989 \pm 0.001$ | $1.002 \pm 0.002$ |
| C = 6.0 | $0.994 \pm 0.001$ | $1.003 \pm 0.002$ |
| C = 9.0 | $0.992 \pm 0.004$ | $1.002 \pm 0.001$ |

# 5 Related Work

## 5.1 Hierarchical Clustering

Agglomerative algorithms iteratively merge clusters based on the shortest distance, building a hierarchy from individual points until converged to a single cluster. Different choices to define the inter-cluster distance give rise to various linkage algorithms [21]. Divisive algorithms proceed in reverse and build hierarchies by splitting a single cluster, containing all points, into smaller and smaller clusters, e.g., by repeatedly applying the $k$-means algorithm [43] or spectral clustering [14].

Recently proposed cost functions, such as the Dasgupta cost [14] and the Tree Sampling Divergence (TSD) [10], have enabled global optimization across the entire hierarchy. While these cost functions remain infeasible to optimize directly, various continuous relaxations have been proposed to allow for gradient-based optimization [11, 34, 9, 48, 30]. These methods then optimize relaxed objectives.

Similar to hierarchical clustering algorithms, TreeGen aims to infer hierarchies and shares the same parametrization for probabilistic hierarchies as Zügner et al. [48] and Kollovieh et al. [30]. However, we focus on a supervised setting and learn a distribution of hierarchies, whereas the former focuses on inferring hierarchies based on heuristics or objective functions.

## 5.2 Jet Clustering

Jet clustering is a well-studied task in high-energy physics. Initially, cone algorithms [5], which select the most energetic particles and group all particles within a cone around them, were employed for this task. However, these algorithms were sensitive to low-energy particles [29]. This limitation led to the development of sequential recombination clustering algorithms, which are instances of agglomerative hierarchical clustering. Different algorithms in this class, such as the Cambridge/Aachen algorithm [15, 46], the $k_T$ algorithm [17], and the anti-$k_T$ algorithm [7], differ in how they combine angular proximity and energy of particles into a distance measure.

Recently, the development of the GINKGO simulator [12] introduced a new class of algorithms that use this simulator to assess the likelihood of splits in the hierarchy, aiming to find the hierarchy with the maximum likelihood [6, 20, 33, 13]. Among those, cluster trellis [33] is most similar to our method as it enables sampling from the exact posterior distribution for small hierarchies. However, our model learns directly from simulator-generated data without requiring likelihood evaluations at inference time, which allows it to be applied with any black-box simulator. Finally, Yang et al. [47] propose a variational-inference method for jet clustering on the GINKGO dataset.

## 5.3 Generative Models

Our work builds on recent advances in generative modeling, particularly diffusion and flow-matching frameworks. Diffusion models add random noise to data and then learn to reverse this diffusion process to generate samples starting from noise [39, 23, 40, 41]. Flow matching generalizes this idea by directly learning an ODE that transforms data to noise via regression of vector fields [32]. Furthermore, Graves et al. [19] introduced BFNs, another related generative framework that maintains and updates a probability distribution during generation in a Bayesian manner rather than acting directly on the sample. Atkinson et al. [1] proposed an ODE-based BFN sampling algorithm that replaces aggregated previous predictions with the most recent prediction, also providing explicit precision control, similar to ours.

Generative models for discrete data are particularly relevant to our approach since hierarchies are discrete structures. One class of such models maintains discreteness throughout the generative process. Various approaches have adapted diffusion models to discrete data [24, 2, 8]. Recent flow matching approaches confine the generative process in continuous space. For example, Dirichlet Flow Matching [42] constrains the process on the probability simplex, while CatFlow [16] removes this constraint, allowing values outside the simplex. Our approach belongs to the former category, as we build upon the BFN framework, transitioning between categorical distributions.

Prior work, such as the junction tree autoencoder [25], modeled trees in an autoregressive fashion for molecule generation. Other approaches used diffusion and flow matching for general graph generation [35, 26, 45, 16]. Unlike these, however, our framework explicitly parametrizes valid hierarchies (rather than distributions over all graphs), substantially reducing the search space.

# 6  Conclusion

In this work, we introduced TreeGen, a novel generative model learning hierarchies from distributions building upon the BFN [19] and BSI [31] framework. By proposing an entropy scheduler, we are able to model categorical variables smoothly across a varying number of classes. The intermediate states of the generation process are probabilistic hierarchies, which we used to derive meaningful features, improving generative performance.

We assessed TreeGen on a high-energy jet simulator benchmark and found that close to 100% of the generated hierarchies adhere to their corresponding physical properties. Moreover, by comparing the likelihoods of conditionally generated hierarchies to those of the ground truth, we demonstrated that our model is able to successfully approximate the posterior distribution.

**Limitations and future work.**  A core component of our generative model is a graph transformer whose computational complexity increases quadratically with the number of nodes. This issue arises from the densely connected probabilistic hierarchies during generation.

To this end, we have evaluated TreeGen on GINKGO [12] as it provides analytical likelihood computations. Immediate extensions include evaluating TreeGen on different simulators [3, 4] and experimental, i.e., real-world jet datasets, where measurement noise and detector effects introduce additional complexity. TreeGen naturally extends to non-binary trees by adjusting the number of internal nodes $n'$ (for binary trees $n - 1$). Potential strategies include sampling $n'$ (as in molecule generation) [45]; predicting $n'$ using a classifier similar to Kerrigan et al. [27]; or setting a larger $n'$ and pruning unused internal nodes, as done in hierarchical clustering [48]. Finally, one could explore alternative hierarchy parameterizations enforcing strict binary-tree constraints or improving both computational efficiency and likelihoods.

## Contributions

MK developed the project idea, derived the theory, implemented the core method, and wrote the manuscript. NF implemented most of the experimental pipeline (data loading, features, backbone, baselines, training, and evaluation) and assisted with the manuscript. FG supervised the neural network architecture design. BS contributed to conceiving the initial idea. SG contributed to the method design and provided overall scientific guidance. All authors discussed results, both theoretical and empirical, and revised the manuscript.

## Acknowledgments

We thank Johanna Sommer for valuable feedback on the idea and Marten Lienen for help with the background section.

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

# A Theory

In the following, we will provide derivations for equations Eqs. (13) to (15). Both theorems build on the theory of Bayesian Flow Networks and summarize the derivations from Graves et al. [19].

**Theorem A.1.** *Let $x \in \{1, \ldots, K\}$ have prior $\boldsymbol{\pi}^{(0)} \in \Delta_{K-1}$. For each update $i = 1, \ldots, N$ we observe $\mathbf{y}^{(i)} = (y_1^{(i)}, \ldots, y_K^{(i)}) \in \mathbb{N}^K$, with total count $m = \sum_k y_k^{(i)}$, sampled from*

$$p(\mathbf{y} \mid x = k, \omega) = \frac{m!}{\prod_j y_j!} \left(\frac{1-\omega}{K}\right)^{m-y_k} \left(\frac{1-\omega}{K} + \omega\right)^{y_k}, \qquad \omega \in (0,1). \tag{20}$$

*Then the posterior after $N$ sequential Bayesian updates is*

$$\boldsymbol{\pi}^{(N)} = \mathrm{softmax}\left(\log \boldsymbol{\pi}^{(0)} + \mathbf{y} \log \pi'\right), \tag{21}$$

*where $\pi' := 1 + \frac{\omega K}{1-\omega}$ and $\mathbf{y}$ the sum of $N$ samples, i.e., $\mathbf{y} := \sum_{i=1}^{N} \mathbf{y}^{(i)}$.*

*Proof.* We prove by induction on $N$.

**Base case ($N = 1$).** By Bayes' rule for a single observation $\mathbf{y}^{(1)}$,

$$\pi_k^{(1)} = p(x = k \mid \mathbf{y}^{(1)}; \omega) \tag{22}$$

$$= \frac{p(\mathbf{y}^{(1)} \mid x = k; \omega) \pi_k^{(0)}}{\sum_{k'=1}^{K} p(\mathbf{y}^{(1)} \mid x = k'; \omega) \pi_{k'}^{(0)}} \tag{23}$$

$$= \frac{\frac{m!}{\prod_{j=1}^{K} y_j!} \left[\frac{1-\omega}{K}\right]^{m-y_k^{(1)}} \left[\frac{1-\omega}{K} + \omega\right]^{y_k^{(1)}} \pi_k^{(0)}}{\sum_{k'=1}^{K} \frac{m!}{\prod_{j=1}^{K} y_j!} \left[\frac{1-\omega}{K}\right]^{m-y_{k'}^{(1)}} \left[\frac{1-\omega}{K} + \omega\right]^{y_{k'}^{(1)}} \pi_{k'}^{(0)}} \tag{24}$$

$$= \frac{\left[\frac{1-\omega}{K}\right]^m \left(1 + \frac{\omega K}{1-\omega}\right)^{y_k^{(1)}} \pi_k^{(0)}}{\left[\frac{1-\omega}{K}\right]^m \sum_{k'=1}^{K} \left(1 + \frac{\omega K}{1-\omega}\right)^{y_{k'}^{(1)}} \pi_{k'}^{(0)}} \tag{25}$$

$$= \frac{\pi'^{y_k^{(1)}} \pi_k^{(0)}}{\sum_{k'=1}^{K} \pi'^{y_{k'}^{(1)}} \pi_{k'}^{(0)}}, \tag{26}$$

$$= \mathrm{softmax}\left(\log \boldsymbol{\pi}^{(0)} + \mathbf{y}^{(1)} \log \pi'\right)_k. \tag{27}$$

Thus, $\boldsymbol{\pi}^{(1)} = \mathrm{softmax}\left(\log \boldsymbol{\pi}^{(0)} + \mathbf{y}^{(1)} \log \pi'\right)$ matching the formula as $\sum_{i=1}^{1} \mathbf{y}^{(i)} = \mathbf{y}^{(1)}$.

**Inductive step.** Assume the statement holds for some $N \geq 1$:

$$\pi_k^{(N)} = \mathrm{softmax}\left(\log \boldsymbol{\pi}^{(0)} + \left[\sum_{i=1}^{N} \mathbf{y}^{(i)}\right] \log \pi'\right)_k.$$

After receiving $\mathbf{y}^{(N+1)}$ we update:

$$\pi_k^{(N+1)} = \frac{(\pi')^{y_k^{(N+1)}} \pi_k^{(N)}}{\sum_j (\pi')^{y_j^{(N+1)}} \pi_j^{(N)}}. \tag{28}$$

We insert the inductive hypothesis for $\pi_k^{(N)}$:

$$\pi_k^{(N+1)} = \frac{(\pi')^{y_k^{(N+1)}} \pi_k^{(N)}}{\sum_j (\pi')^{y_j^{(N+1)}} \pi_j^{(N)}} \tag{29}$$

$$= \frac{(\pi')^{y_k^{(N+1)}} \dfrac{\exp\big(\log \pi_k^{(0)} + \log \pi' \sum_{i=1}^N y_k^{(i)}\big)}{\sum_\ell \exp\big(\log \pi_\ell^{(0)} + \log \pi' \sum_{i=1}^N y_\ell^{(i)}\big)}}{\sum_j (\pi')^{y_j^{(N+1)}} \dfrac{\exp\big(\log \pi_j^{(0)} + \log \pi' \sum_{i=1}^N y_j^{(i)}\big)}{\sum_\ell \exp\big(\log \pi_\ell^{(0)} + \log \pi' \sum_{i=1}^N y_\ell^{(i)}\big)}} \tag{30}$$

$$= \frac{\exp\big(\log \pi_k^{(0)} + \log \pi' \big[\sum_{i=1}^N y_k^{(i)} + y_k^{(N+1)}\big]\big)}{\sum_j \exp\big(\log \pi_j^{(0)} + \log \pi' \big[\sum_{i=1}^N y_j^{(i)} + y_j^{(N+1)}\big]\big)} \tag{31}$$

$$= \frac{\exp\big(\log \pi_k^{(0)} + \log \pi' \sum_{i=1}^{N+1} y_k^{(i)}\big)}{\sum_j \exp\big(\log \pi_j^{(0)} + \log \pi' \sum_{i=1}^{N+1} y_j^{(i)}\big)}. \tag{32}$$

Because $\sum_{i=1}^N y_k^{(i)} + y_k^{(N+1)} = \sum_{i=1}^{N+1} y_k^{(i)}$, the numerator and denominator are exactly the softmax with the sum taken to $N+1$. Hence

$$\pi_k^{(N+1)} = \mathrm{softmax}\Big(\log \boldsymbol{\pi}^{(0)} + \Big[\sum_{i=1}^{N+1} \mathbf{y}^{(i)}\Big] \log \pi'\Big)_k, \tag{33}$$

Thus, $\boldsymbol{\pi}^{(N+1)} = \mathrm{softmax}\Big(\log \boldsymbol{\pi}^{(0)} + \Big[\sum_{i=1}^{N+1} \mathbf{y}^{(i)}\Big] \log \pi'\Big)$ completing the induction. $\qquad\square$

**Theorem A.2.** *Fix $K \geq 2$ and $\omega \in (0,1)$. Let $x \in \{1,\dots,K\}$ have prior $\boldsymbol{\pi}_0 \in \Delta_{K-1}$. At each time step $t \in \mathbb{N}$ an observation $\mathbf{y}^{(t)} \in \mathbb{N}^K$ with $\sum_k y_k^{(t)} = m$ is drawn from*

$$p(\mathbf{y}^{(t)} \mid x = k, \omega) = \frac{m!}{\prod_j y_j^{(t)}!} \left(\frac{1-\omega}{K}\right)^{m - y_k^{(t)}} \left(\frac{1-\omega}{K} + \omega\right)^{y_k^{(t)}}.$$

*Define $\pi' := 1 + \dfrac{\omega K}{1 - \omega}$ and $\Delta \alpha_t := \alpha_t - \alpha_{t-1} = m\omega^2$.*

*Then the following posterior approximations hold for $m \to \infty$:*

1. ***Single–step update.*** *Given $\boldsymbol{\pi}_{t-1}$, $\boldsymbol{\pi}_t$ can be approximated using a Gaussian random vector:*

$$\boldsymbol{\pi}_t \approx \mathrm{softmax}\big(\log \boldsymbol{\pi}_{t-1} + \mathbf{z}_t\big), \qquad \mathbf{z}_t \sim \mathcal{N}\big(\Delta \alpha_t K \mathbf{e}_x, \Delta \alpha_t K \mathbf{I}\big). \tag{34}$$

2. ***Aggregated update.*** *$\boldsymbol{\pi}_t$ can be approximated using a Gaussian random vector directly:*

$$\boldsymbol{\pi}_t \approx \mathrm{softmax}(\mathbf{z}_t), \ \mathbf{z}_t \sim \mathcal{N}(\alpha_t K \mathbf{e}_x, \alpha_t K \mathbf{I}). \tag{15}$$

*Proof.* **(i) Single step.** Conditioned on $x$, the vector $\mathbf{y}^{(t)}$ is multinomial with parameters $m$ and probabilities

$$\boldsymbol{\theta} = \omega \, \mathbf{e}_x + (1 - \omega) \tfrac{1}{K} \mathbf{1}. \tag{35}$$

The Central Limit Theorem yields:

$$\mathbf{y}^{(t)} \xrightarrow[m\to\infty]{d} \mathcal{N}\big(m\omega \mathbf{e}_x + (1-\omega)\tfrac{1}{K}\mathbf{1}, m \, \mathrm{diag}(\boldsymbol{\theta}) - \boldsymbol{\theta}\boldsymbol{\theta}^\top\big). \tag{36}$$

By applying a mean-field approximation $\mathrm{diag}(\boldsymbol{\theta}) - \boldsymbol{\theta}\boldsymbol{\theta}^\top \approx \frac{1}{K}\mathbf{I}$ and noting that $\log \pi' \approx \omega K$ since $\omega \to 0$, we obtain:

$$\mathbf{y}^{(t)} \log \pi' \xrightarrow[m \to \infty]{d} \mathcal{N}\big(m\omega^2 K\mathbf{e}_x + \omega(1 - \omega)\mathbf{1}, m\omega^2 K\mathbf{I}\big). \tag{37}$$

Since the softmax operation is shift-invariant, we can drop the constant $\omega(1 - \omega)\mathbf{1}$ and define $\mathbf{z}_t \sim \mathcal{N}\big(m\omega^2 K\mathbf{e}_x, m\omega^2 K\mathbf{I}\big)$. Plugged into the update, we obtain $\boldsymbol{\pi}_t = \mathrm{softmax}(\log \boldsymbol{\pi}_{t-1} + \mathbf{z}_t)$ matching Eq. (14).

**(ii) Aggregation.** The vectors $\mathbf{z}_1, \ldots, \mathbf{z}_t$ are conditionally i.i.d. Gaussians; their sum is Gaussian with mean $\sum_{s=1}^{t} \Delta\alpha_t K\mathbf{e}_x = \alpha_t K\mathbf{e}_x$ and covariance $\sum_{s=1}^{t} \Delta\alpha_t K\mathbf{I} = \alpha_t K\mathbf{I}$. Because a constant vector can be subtracted inside the softmax, the factor $\log \boldsymbol{\pi}_0$ drops out, yielding $\boldsymbol{\pi}_t = \mathrm{softmax}(\mathbf{z}_t)$ with the stated distribution. $\qquad\square$

## A.1 Differences to BFNs.

While there is a different theoretical perspective between TreeGen and BFNs, both models build on the same Bayesian update. The key differences mostly occur in the sampling and training procedure. TreeGen performs the following update during sampling:

$$\hat{\boldsymbol{\pi}} = f_\theta(\boldsymbol{\pi}_{t-1}, t - 1) \tag{38}$$

$$\mathbf{z}_t \sim \mathcal{N}\big(\alpha_t K\hat{\boldsymbol{\pi}}, \alpha_t K\mathbf{I}\big) \tag{39}$$

$$\boldsymbol{\pi}_t = \mathrm{softmax}(\mathbf{z}_t). \tag{40}$$

In contrast, BFNs perform:

$$\hat{\boldsymbol{\pi}} = f_\theta(\boldsymbol{\pi}_{t-1}, t - 1) \tag{41}$$

$$\mathbf{z}_t \sim \mathcal{N}\big(\tfrac{d\alpha_t}{dt} K\hat{\boldsymbol{\pi}}, \tfrac{d\alpha_t}{dt} K\mathbf{I}\big) \tag{42}$$

$$\boldsymbol{\pi}_t = \mathrm{softmax}(\boldsymbol{\pi}_{t-1} + \mathbf{z}_t). \tag{43}$$

Furthermore, BFNs optimize their framework using the following objective:

$$\mathcal{L}(\theta) = \mathbb{E}_{x \sim \mathcal{D}, t \sim \mathcal{U}(0,1), \boldsymbol{\pi}_t \sim p_t(\boldsymbol{\pi}_t | \mathbf{x})} \left[ \tfrac{d\alpha_t}{2dt} \| f_\theta(\boldsymbol{\pi}_t, t) - e_x \|_2^2 \right], \tag{44}$$

while TreeGen use the loss presented in Sec. 3.

# B  Experiment Details

## B.1  Hyperparameters

We provide an overview of the training and model hyperparameters in Tab. 3.

Table 3: Hyperparameters of TreeGen.

| | Training | | | | Sampling | Graph-Transformer | | | Upscaler | | Prediction Head |
| | LR | Epochs | Grad. Clip | EMA | Steps | Layers | Dim ($h$) | Heads | Node Layers | Edge Layers | Layers |
|---|---|---|---|---|---|---|---|---|---|---|---|
| Value | $10^{-4}$ | 50 | 0.5 | 0.9999 | 1000 | 4 | 64 | 2 | 1 | 3 | 2 |

All experiments are conducted on A100 GPUs.

## B.2  Datasets

We simulate five jet-shower datasets using the GINKGO generator [12]. Three contain QCD jets (QCD-S, QCD-M, QCD-L), and two describe W-boson jets (W-S, W-M). All sets share a common decay rate $\lambda$ and differ only in the shower cut-off $\Delta_{\text{cut}} \in \{4.0^2,\, 1.1^2,\, 0.6^2\}$, which controls the tree sizes: up to 19 nodes for S, 59 for M, and 99 for L. Each dataset contains $10^5$ binary hierarchies, split into 98,000 training examples and 1,000 each for validation and test. Fig. 8 plots the leaf-count distributions of the three QCD samples, illustrating how the cut-off controls the final hierarchy size. Inside every hierarchy, the nodes, i.e., an energy-momentum vector specifies particles:

$$\boldsymbol{p}^{\mu} = \big(E,\ p_x,\ p_y,\ p_z\big), \tag{45}$$

which contains the energy $E$ and momentum $(p_x, p_y, p_z)$. In our setting, we only observe the leaves and infer the hierarchy, i.e., the structure of internal nodes.

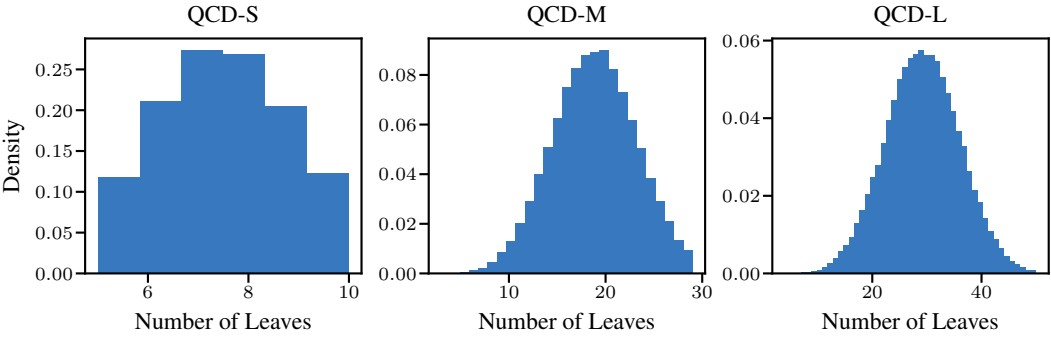

Figure 8: Histogram of the number of leaves for the three different QCD datasets.

In our setup, the tree corresponds to a splitting process. A parent node splits into its children. The root corresponds to the initial particle, while the leaves represent the observed particles. We aim to reconstruct the splitting process, i.e., find the edges of the tree by inferring the parent–child edges given the leaves.

## B.3  Architecture

The input to our neural network is a probabilistic hierarchy with $n + n'$ nodes, representing leaves and internal nodes, i.e., $\mathcal{T} \in [0,1]^{(n+n') \times (n+n')}$. Our architecture consists of three components: (*i*) the feature generation (discussed in App. B.4) , (*ii*) a graph transformer, and (*iii*) a prediction head.

**Graph Transformer.**   Given $h$-dimensional node and edge features, denoted as $\boldsymbol{X} \in \mathbb{R}^{(n+n') \times h}$ and $\boldsymbol{E} \in \mathbb{R}^{m \times h}$, respectively, we iteratively update node features $\boldsymbol{X}^{(l)}$ at layer $l$ given $\boldsymbol{X}^{(l-1)}$, $\boldsymbol{E}$, and the edges of the hierarchy as adjacency matrix $\mathcal{A} \in \{0,1\}^{(n+n') \times (n+n')}$. We use the graph transformer proposed by Shi et al. [38] as our backbone with four layers, two attention heads, and a hidden dimension of 64 (Tab. 3).

**Prediction head.**    Our goal is to predict parent probabilities, i.e., edge probabilities from the node representations of the graph-transformer. Therefore, we apply a prediction head to each child-parent pair $(v_i, z_i)$ to obtain a logit for the corresponding edge and apply a softmax over all outgoing edges of the child.

The prediction head concatenates the latent representations of the child and the parent together with the corresponding edge features. These are then passed through a small MLP to obtain the logits as depicted in Fig. 9.

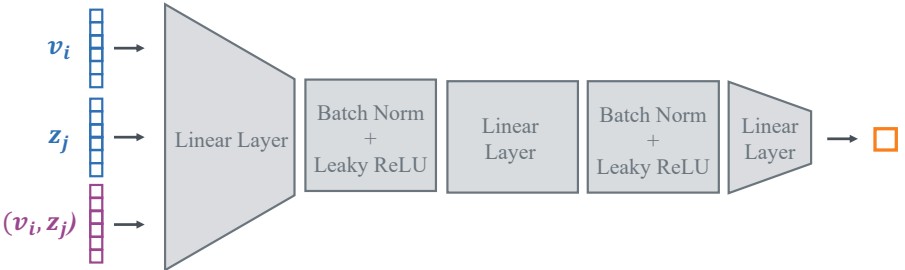

Figure 9: **Prediction head.** The prediction head processes the representations of child-parent pairs $(v_i, z_j)$, together with their corresponding edge features, and predicts single logit.

## B.4    Feature Generation

To model probabilistic hierarchies effectively, we extract a diverse set of features. We first introduce features encoding structural and uncertainty-related information such as node type, entropy, and ancestor probabilities (App. B.4.1). We then present physically-inspired features, which capture domain-specific properties like momentum, rapidity, and invariant mass (App. B.4.2). Finally, we project raw features into the hidden space used by the network (App. B.4.3).

### B.4.1    Features for Probabilistic Hierarchies

In this section, we present node and edge features that can be derived from probabilistic hierarchies and aid the network in obtaining global features. We use the equations and sampling procedure from Zügner et al. [48].

**Node Type.**    If nodes have children, they are categorized as internal nodes; otherwise, as leaves. We represent this with a binary feature. Furthermore, we introduce a binary feature identifying the root, as it is the only node without a parent.

**Node Entropy.**    We model each node's parent assignment as a categorical distribution over its candidate parents and derive two features: (i) the number of candidate parents and (ii) a normalized entropy quantifying uncertainty. Specifically, we compute:

$$\mathcal{H}(v_i) = -\frac{1}{\log(n')} \sum_{j=1}^{n'} \boldsymbol{A}_{ij} \log \boldsymbol{A}_{ij}, \tag{46}$$

$$\mathcal{H}(z_i) = -\frac{1}{\log\big(n' - (i+1)\big)} \sum_{j=i+1}^{n'} \boldsymbol{B}_{ij} \log \boldsymbol{B}_{ij}. \tag{47}$$

The logarithmic normalizers match the number of candidate parents in each case, scaling the entropy to $[0, 1]$.

**Expected Children.**    For each internal node $z_k$, we include the expected number of children under the tree sampling procedure $(\hat{\boldsymbol{A}}, \hat{\boldsymbol{B}}) \sim p_{\boldsymbol{A}, \boldsymbol{B}}$ as a feature. Under $p_{\boldsymbol{A}, \boldsymbol{B}}$, this equals the total

probability that $z_k$ is chosen as the parent of each leaf and each earlier internal node $j < k$:

$$\mathbb{E}\left[n_{\mathrm{child}}(z_k)\right] = \sum_{i=1}^{n} p(z_k \mid v_i) + \sum_{j=1}^{k-1} p(z_k \mid z_j) \tag{48}$$

$$= \sum_{i=1}^{n} \boldsymbol{A}_{ik} + \sum_{j=1}^{k-1} \boldsymbol{B}_{jk}, \tag{49}$$

where $A_i$ and $B_j$ denote the parent of leaf $v_i$ and internal node $z_j$, respectively. The equalities follow from the linearity of expectation and $\mathbb{E}[\mathbb{I}[X = k]] = \Pr(X = k)$.

**Sibling Probability.** We encode the sibling relation, i.e., whether two nodes share the same parent, by its probability under the tree-sampling procedure as edge features. For leaves $v_i$ and $v_j$, we compute the probability by:

$$p\left(v_i \text{ and } v_j \text{ are siblings}\right) = \sum_{k=1}^{n'} p(z_k \mid v_i, z_k \mid v_j) \tag{50}$$

$$= \sum_{k=1}^{n'} p(z_k \mid v_i)\, p(z_k \mid v_j) \tag{51}$$

$$= \sum_{k=1}^{n'} \boldsymbol{A}_{ik} \boldsymbol{A}_{jk}, \tag{52}$$

where we used the independence of parent draws in the sampling procedure. For internal nodes, we compute this quantity using $\boldsymbol{B}$, summing only over feasible parents.

**Ancestor Features.** We use two ancestor-based features. (i) *Ancestor probability* as an edge feature for an internal node $z_k$ and a node $v_i$, let $P_{ik}^{\mathrm{anc}} := p(z_k \in \mathrm{anc}(v_i))$ denote the probability—under the tree-sampling procedure—that $z_k$ is an ancestor of $v_i$. (ii) *Expected number of ancestors* as a node feature, which equals the node's expected depth.

For a leaf $v_i$, we compute the expected number of ancestors as:

$$\mathbb{E}[n_{\mathrm{anc}}(v_i)] = \mathbb{E}\left[\sum_{k=1}^{n'} \mathbb{I}[z_k \in \mathrm{anc}(v_i)]\right] \tag{53}$$

$$= \sum_{k=1}^{n'} p(z_k \in \mathrm{anc}(v_i)) \tag{54}$$

$$= \sum_{k=1}^{n'} P_{ik}^{\mathrm{anc}}. \tag{55}$$

The computation for internal nodes follows analogously, using the matrix $\tilde{P}^{\mathrm{anc}}$.

For a pair of nodes, we also compute the (expected) number of *shared* ancestors as a similarity measure. As we cannot derive an exact expression due to the dependence between the ancestor probabilities $p(z_k \in \mathrm{anc}(v_i))$ and $p(z_k \in \mathrm{anc}(v_j))$, we use an independence approximation:

$$\mathbb{E}[n_{\mathrm{anc}}(v_i, v_j)] \approx \sum_{k=1}^{n'} P_{ik}^{\mathrm{anc}} P_{jk}^{\mathrm{anc}}. \tag{56}$$

The computation of $P^{\mathrm{anc}}$ and $\tilde{P}^{\mathrm{anc}}$ follows Zügner et al. [48].

**Hierarchy size.** As the hierarchies can vary in size within the datasets, we include the hierarchy size, i.e., the number of nodes, as a feature.

**Time embeddings.** In addition to the probabilistic hierarchies and their derived features, we include the time $t \in [0, 1]$ as input to the network. We use a 10-dimensional sinusoidal positional encoding as done in the transformer architecture [44]. We concatenate these embeddings to the node features.

### B.4.2 Physically Inspired Features

In addition to the task-agnostic features described above, we include physically inspired features tailored to jet clustering. Each node in the hierarchy corresponds to a particle and is described by its four-momentum vector

$$\boldsymbol{p}^{\mu} = (E, p_x, p_y, p_z),\tag{57}$$

which serves as the basis for our physics-derived features. The momentum $\vec{p} = (p_x, p_y, p_z)$ is defined in three spatial directions: the $z$-axis is aligned with the beam (beamline), and the plane spanned by the two axes orthogonal to $z$ is the transverse plane.

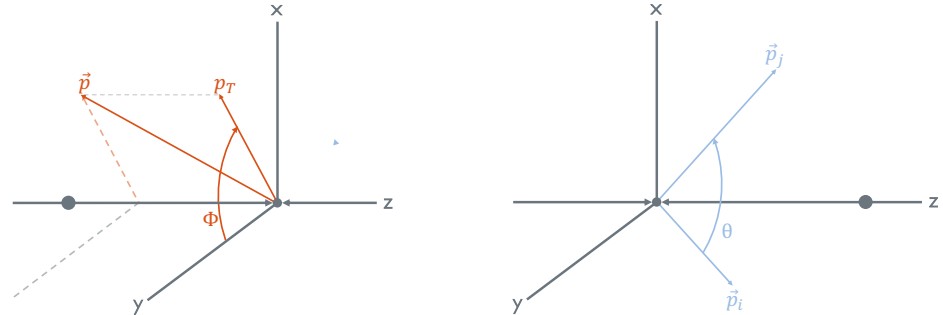

Figure 10: Illustration of the transverse momentum $p_T$ and the azimuth $\phi$ (*left*), and the angle $\theta$ between the momentum vectors of two particles (*right*).

**Transverse Momentum.** The transverse plane is crucial for analyzing the dynamics of particle interactions and scatterings. A key quantity in this context is the transverse momentum:

$$p_T := \sqrt{p_x^2 + p_y^2},\tag{58}$$

which we include as a node feature.

**Angular Features.** Following the anti-$k_T$ algorithm, we compute the azimuth $\phi$ and the rapidity $y$, which define the pairwise distance measure used by the algorithm. The azimuth $\phi$ is measured in the transverse plane and is computed as:

$$\phi := \arctan\left(\frac{p_x}{p_y}\right).\tag{59}$$

The rapidity $y$ is related to the angle of a particle relative to the beamline. It is especially useful in high-energy physics as it remains approximately invariant under boosts along the beam axis:

$$y := \frac{1}{2}\log\left(\frac{E + p_z}{E - p_z}\right).\tag{60}$$

We include $\phi$ and $y$ as node features and, analogously to the anti-$k_T$ algorithm, use their differences as edge features:

$$\Delta\phi = \phi_i - \phi_j \qquad \text{and} \qquad \Delta y = y_i - y_j\tag{61}$$

As an additional edge feature, we use the angle $\theta$, which accounts for all momentum components and is given by:

$$\theta = \arccos\left(\frac{\vec{p}_i^T \vec{p}_j}{\|\vec{p}_i\|_2 \|\vec{p}_j\|_2}\right),\tag{62}$$

where $\vec{p} = (p_x, p_y, p_z)$ is the momentum vector of a particle. We illustrate the angles $\phi$ and $\theta$ in Fig. 10.

**Invariant Mass.** Finally, we use the squared invariant mass, which represents the intrinsic mass of a particle independent of its motion or reference frame:

$$\Delta = E^2 - (p_x^2 + p_y^2 + p_z^2). \tag{63}$$

This property plays a crucial role in the splits within our hierarchies. As discussed in Section B.2, the invariant mass of a parent particle must be greater than the sum of the invariant masses of its children. Hence, for any internal node $z_k$ with children $v_i$ and $z_j$:

$$\sqrt{\Delta_{z_k}} \geq \sqrt{\Delta_{v_i}} + \sqrt{\Delta_{z_j}}. \tag{64}$$

In the probabilistic setting, we relax this condition to

$$\sqrt{\Delta_{z_k}} \geq \sum_{i=1}^{n} \boldsymbol{A}_{ik} \sqrt{\Delta_{v_i}} + \sum_{j=1}^{k-1} \boldsymbol{B}_{jk} \sqrt{\Delta_{z_j}}. \tag{65}$$

We use the difference between the invariant mass of the parent node and the weighted sum of the invariant masses of its children as a node feature. Additionally, we include binary variables whether $\Delta_{z_k}$ exceeds the cut thresholds and whether it's larger than 0.

We also derive edge features from this constraint. For each pair of nodes, we compute the energy-momentum vector of their potential parent and calculate the difference between the invariant mass of this potential parent and the sum of the invariant masses of the two children. This edge feature effectively quantifies the plausibility of the nodes being siblings.

### B.4.3 Feature Upscaling

To align inputs with the hidden size $h$ and stabilize training, we normalize node features with batch normalization, apply a linear layer, and then a LeakyReLU. Because the backbone does not update edge features, we process them using three successive blocks. The upscaling block is shown in Fig. 11. These serve as input to our backbone (see App. B.3).

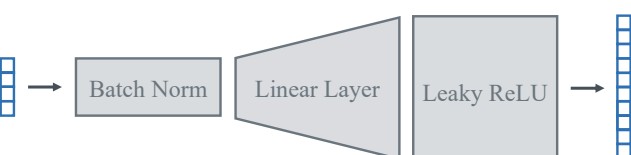

Figure 11: **Feature Upscaler.** The feature upscaler processes the input features and maps them to the hidden dimension $h$.

### B.5 Baselines

In the following, we give a short description of the baselines. Both generative models operate on probabilistic hierarchies, just like TreeGen. We keep architecture, features, training, and sampling parameters consistent.

**Bayesian Flow Networks (BFNs).** The most related baselines are standard BFNs. We discuss details and differences to our model in App. A. For the implementation, we keep it as similar as possible to TreeGen. While we use our proposed entropy scheduler as explained in Sec. 3, we perform training and sampling as described by Graves et al. [19].

**CatFlow.** CatFlow [16] is a flow-matching model that evolves samples on the probability simplex by integrating an ODE whose vector field is parametrized by a neural network. Training uses a cross-entropy loss to predict the ground-truth hierarchy from noisy ones, similar to TreeGen. Two key differences remain: *(i)* CatFlow's trajectories are not constrained to stay on the simplex during integration. We use a Gaussian prior with mean and standard deviation of 0.25, clipped at $10^{-4}$ to enable the computation of all features, and *(ii)* the generative process is deterministic once the prior sample is fixed.

**anti-$k_t$ jet clustering algorithm.** The anti-$k_t$ is a well-established algorithm [7]. It iteratively merges the two entities, i.e., particles or pseudojets, with the smallest pairwise distances $d_{ij}$. The distances are computed based on the transverse momentum, azimuth angle, and rapidity (see App. B.4).

## B.6  Metrics

We report two complementary, physics-aware metrics.

**Valid Hierarchy Percentage (Valid Fraction).** A hierarchy is *valid* if it satisfies three conditions:

   **(i)** it is a binary decay tree

  **(ii)** at every internal node the invariant mass exceeds those of the sum of the children

  **(iii)** the squared invariant mass of each internal particle exceeds the dataset-specific cut-off.

For every generated hierarchy, we check these conditions and return the percentage that are valid.

**Ratio of Log-Likelihoods (LLH Fraction).** Physical plausibility does not guarantee that samples follow the data distribution. Leveraging the Ginkgo simulator, we compute the log-likelihood of each generated hierarchy ($\ell_{\text{gen}}$) and of its ground-truth counterpart ($\ell_{\text{true}}$). We report the average ratio $\ell_{\text{true}}/\ell_{\text{gen}}$ over the test set. A value close to one indicates that the generative model aligns with the target distribution.

# C  Additional Results

## C.1  Feature ablation

In the following, we ablate core features discussed in App. B.4 on QCD-S. We start with a base model that neither uses features from the ancestor nor physical properties. The results are shown in Tab. 4.

Table 4: Ablation of ancestor and physically inspired features.

| Model | Valid Frac. ($\uparrow$) | LLH Frac. |
|---|---|---|
| Base model | $0.563 \pm 0.004$ | $0.937 \pm 0.002$ |
| + Ancestor Features | $0.564 \pm 0.001$ | $0.939 \pm 0.001$ |
| + 4-Momentum + Angular Features | $0.645 \pm 0.021$ | $0.982 \pm 0.001$ |
| + Invariant Mass | $0.997 \pm 0.001$ | $1.003 \pm 0.002$ |

We observe that the base model only provides 56.3% of valid trees. While including the ancestor features only provides marginal improvements, the physically inspired features yield major improvements.

## C.2  Comparison to clustering algorithms.

In Tab. 5, we compare TreeGen to the CA [15, 46], $k_t$ [17], and anti-$k_t$ [7] jet clustering algorithm. As we observe, TreeGen consistently outperforms the traditional clustering algorithms in terms of

Table 5: Comparison between clustering algorithms and TreeGen. Best scores in **bold**.

| Dataset | CA | | $k_t$ | | anti-$k_t$ | | TreeGen | |
|---|---|---|---|---|---|---|---|---|
| | Valid ($\uparrow$) | LLH | Valid ($\uparrow$) | LLH | Valid ($\uparrow$) | LLH | Valid ($\uparrow$) | LLH |
| QCD-S | 0.416 | 0.989 | 0.302 | **1.029** | 0.840 | 0.873 | **0.997** $\pm 0.001$ | 1.003 $\pm 0.002$ |
| QCD-M | 0.088 | 0.987 | 0.050 | **1.050** | 0.552 | 0.752 | **0.977** $\pm 0.010$ | 0.994 $\pm 0.002$ |
| QCD-L | 0.025 | 0.978 | 0.008 | **1.065** | 0.440 | 0.636 | **0.943** $\pm 0.016$ | 0.975 $\pm 0.001$ |

valid hierarchies. In terms of likelihood, the $k_t$ algorithm achieves slightly higher scores.

