# OpenReview forum: "TreeGen: A Bayesian Generative Model for Hierarchies"
_NeurIPS.cc/2025/Conference — NeurIPS 2025 poster_

### Official Review · Reviewer_uAVU · 2025-06-30

**Clarity:** 3
**Significance:** 2
**Originality:** 2
**Rating:** 4
**Confidence:** 2

**Summary:**

The paper presents TreeGen, a novel generative model that extends Bayesian Flow Networks (BFNs) to capture distributions over hierarchical data. In addition to the model, the authors propose a new entropy scheduling strategy for BFNs that performs well across various structures without requiring hyperparameter tuning. TreeGen is designed to solve jet clustering problems, and the experimental results demonstrate that it outperforms existing methods across multiple datasets.

**Questions:**

1. As someone not familiar with jet clustering, I am somewhat confused by the underlying prediction problem. If I understand it correctly, each node in the tree corresponds to an individual particle and the goal is to learn a graph neural network that generalizes across different particles, i.e., predict the probability of each edge in the tree given the representations of the parent node, the child node, and the edge itself. Is that assessment correct?
2. In Section 3, each node in the tree is treated as a random variable which is observed multiple times. Does that mean that each node represents a type of particle rather than individual particles, and that is why we can collect repeated observations for each?
3. Do all trees in the training and test datasets have to share the same structure? That is, would be possible to generalize across different structures provided we have good features for all nodes?

**Ethical Concerns:**

["NO or VERY MINOR ethics concerns only"]

**Final Justification:**

I think the proposed methods are novel and interesting enough to justify acceptance. However, as I mentioned in my review, I would have liked to see the methods applied in other domains or at least presented in a more general fashion, as I see jet clustering as a somewhat niche application within the Neurips community. This is the reason I maintain my score as a 4 for now. Of course, as someone not entirely familiar with the area of machine learning for physics, I am happy to raise my score to a 5 if I have got the wrong impression about the relevance of the topic here.

**Limitations:**

Yes

**Paper Formatting Concerns:**

No concerns.

**Quality:**

3

**Strengths And Weaknesses:**

### Strengths
-----------
- The main method of the paper, TreeGen, extends Bayesian Flow Networks (BFNs) to hierarchical data and is novel, to the best of my knowledge.
- The experiments are well designed and the results indicate TreeGen outperforms CatFLow and BFNs on a number of jet clustering datasets.
- The mathematical development is clear and sound.
- The paper is very well written and organized.

### Weaknesses
------------
I should mention that I am not familiar with the literature on jet clustering, so I am not in a position to assess whether it is a problem of particular relevance to the NeurIPS community. From my perspective, the paper seems more focused on addressing the jet clustering task itself than on introducing a broadly applicable machine learning method. If jet clustering is already well-established as a relevant topic within the community, then this focus makes perfect sense. However, if that is not the case, it might be helpful for the authors to make the paper more approachable to a broader machine learning audience. For example, by providing a clearer explanation of the problem (see questions below) and highlighting how the proposed approach or insights could be useful in other machine learning contexts.

---

> ### Author Rebuttal · Authors · 2025-07-30
>
> We thank the reviewer for their valuable feedback. In the following, we address their comments.
>
> **Comment:** As someone not familiar with jet clustering, I am somewhat confused by the underlying prediction problem. If I understand it correctly, each node in the tree corresponds to an individual particle and the goal is to learn a graph neural network that generalizes across different particles, i.e., predict the probability of each edge in the tree given the representations of the parent node, the child node, and the edge itself. Is that assessment correct?
> **Response:** The assessment is correct. Each node represents a particle, and our network predicts the probability of each possible parent-child edge based on the features of the particles and current edges in the tree. The edges themselves encode the splitting process, i.e., the parent particle splits into the two child particles.
>
> **Comment:** In Section 3, each node in the tree is treated as a random variable which is observed multiple times. Does that mean that each node represents a type of particle rather than individual particles, and that is why we can collect repeated observations for each?
> **Response:** The tree corresponds to a splitting process. A parent node splits into its children. The root corresponds to the initial particle, while the leaves represent the observed particles. We collect repeated observations by making predictions with the neural network, allowing us to refine our beliefs. We added a paragraph in the manuscript clarifying how particles correspond to nodes in our setup.
>
> **Comment:** Do all trees in the training and test datasets have to share the same structure? That is, would be possible to generalize across different structures provided we have good features for all nodes?
> **Response:** All trees in the datasets are binary trees, i.e., have binary branches but vary in size, shape, and features (4 momentum). We will clarify this in the experiments section.
>
> Finally, we would like to highlight the importance of our work to different audiences:
> 1. High-energy physics, where TreeGen improves over traditional jet-clustering algorithms.
> 2. Hierarchical modelling, since our method is not limited to jets but can be applied to any hierarchical data.
> 3. Generative modelling, as our derivation simplifies BFNs and allows for a broader range of modelling choices with substantial empirical improvements.
>
> We thank the reviewer again for their feedback and have improved the clarity based on their comments by introducing an additional paragraph in the updated manuscript describing the jet clustering task. We are happy to address any remaining concerns.

---

> > ### Comment · Reviewer_uAVU · 2025-08-01
> >
> > Thanks for engaging and for solving my questions.
> >
> > Regarding the trees, would it be possible to extend the method to arbitrary tree structures or are binary trees enough, e.g. via the introduction of latent nodes (variables) in the tree?
> >
> > Regarding the audience of the paper, I understand the method applies more broadly than jet clustering. My point was that the paper was written with jet clustering in mind throughout, from the title to the experimental design, and I am not convinced this was the best strategy to attract the different audiences you list.

---

> > > ### Author Response · Authors · 2025-08-02
> > >
> > > Thank you for your response.
> > >
> > > TreeGen extends to arbitrary tree structures as our parametrization allows for directed acyclic graphs. The only adjustment necessary is choosing the number of internal nodes $n'$, which for binary trees is $n-1$, but can vary for arbitrary branching factors.
> > >
> > > While introducing latent nodes in the tree is possible, we see various approaches done in previous work:
> > >
> > > 1. Sample $n'$ randomly, similar to previous works, which samples the number of atoms for molecule generation [1].
> > > 2. Predict $n'$ via a neural network on conditions, which has been explored in the context of temporal point processes [2]. In our case, this could be conditioned on, for example, the leaf features.
> > > 3. Finally, one could choose a higher $n'$ and remove unused internal nodes, similar to previous approaches in a clustering context [3].
> > >
> > > Regarding the intended audience, we agree that the jet-clustering task dominates in our paper. Our goal was to motivate and evaluate TreeGen on an important, concrete task, while keeping the methodology and background section accessible to a broader audience. We will revise the introduction to clarify that our method is not limited to high-energy physics, and update our future work paragraph accordingly. Furthermore, we welcome further suggestions to make our work more accessible to a broader audience.
> > >
> > > [1] **Vignac, Clement, Igor Krawczuk, Antoine Siraudin, Bohan Wang, Volkan Cevher, and Pascal Frossard.** "Digress: Discrete denoising diffusion for graph generation." arXiv preprint arXiv:2209.14734 (2022).
> > > [2] **Kerrigan, Gavin, Kai Nelson, and Padhraic Smyth.** "EventFlow: Forecasting Temporal Point Processes with Flow Matching." arXiv preprint arXiv:2410.07430 (2024).
> > > [3] **Zügner, Daniel, Bertrand Charpentier, Morgane Ayle, Sascha Geringer, and Stephan Günnemann.** "End-to-end learning of probabilistic hierarchies on graphs." In International Conference on Learning Representations. 2021.

---

> > > > ### Comment · Reviewer_uAVU · 2025-08-07
> > > >
> > > > Thank you go for the clarifications on how to expand the method to arbitrary trees. This is an interesting point that adds to the paper.
> > > >
> > > > Regarding the presentation, personally, I would only have mentioned jet clustering in the experiments section to make it clear that the proposed method is general as you say. I would also have included experiments in other domains to demonstrate the flexibility of the method.
> > > >
> > > > That said, as I mentioned in my original review, this hinges on a subjective opinion about the interest in and impact of jet clustering in the Neurips community. If the other reviewers are happy with how the paper is presented or convinced of the relevance of jet clustering in general, I am happy to have the paper accepted.

---

> > > > > ### Author Response · Authors · 2025-08-08
> > > > >
> > > > > Thank you for your response.
> > > > >
> > > > > We will include these clarifications in the updated manuscript.

---

> ### Author Response · Authors · 2025-08-06
>
> Dear reviewer,
>
> We hope our additional clarifications have addressed your comments satisfactorily. We remain fully available to provide clarifications or answer any further questions they may have.
>
> Again, thank you for your efforts and thoughtful feedback.

---

### Official Review · Reviewer_tb6N · 2025-07-01

**Clarity:** 4
**Significance:** 3
**Originality:** 3
**Rating:** 5
**Confidence:** 5

**Summary:**

Jet clustering in particle physics is an algorithmic process that links low energy hadronic detector outputs to high energy partonic states. While the standard usage relies on unsupervised clustering methods which is useful for discovery and is also agnostic to the required analysis, it is known that different signals require bespoke setting of parameters such as jet radius.  These are often tested on simulated showered events in order to extract physically meaningful quantities from these parameter-dependent jets.  This paper seeks to extract more meaningful patterns by learning a generative model for probabilistic trees to represent stochastically generated showers.   They base this on prior work on flow networks with Bayesian conditional updates with an entropy based accumulated signal, which is a scaling of a Gaussian random variable that approximates a discrete sum over categorical variables.  The conditional update over the discrete random variable is extended to the hierarchical setting required for tree generation.  The evaluation is done using Ginkgo datasets for QCD and W boson showers. For the small datasets a direct comparison of the posterior distributions is feasible and shown.  Per generated tree likelihoods are reported as well as maximum a posteriori tree likelihoods, a property enabled by the generative nature of the learned tree models.

**Questions:**

At the risk of repeating the comments from above, here are some questions.

(1) Could you add comparisons with Cambridge-Aachen and the $k_T$ algorithms?

(2) From the numbers in Table 3, it is clear that the invariant mass plays a key role in outperforming anti-$k_T$.  Could you provide some explanations as to why this might be the case?

(3) Why do the angular features not guarantee l. 968-9 (ii-iii) is satisfied, whereas an unsupervised method appears to do so by virtue of the agglomerative step?

(4) Does this not invalidate the method as a reliable probe of jets that are typically extracted via anti-$k_T$ at the LHC and later processed via C-A and $k_T$ algorithms for substructure and tagging studies, and for pileup removal?

(5) Please provide potentially physically meaningful insights that this, otherwise elegant, approach might help extract.  Given the concerns I have raised, being able to appreciate what this way of processing jets might bring to typical tasks undertaken would enhance my assessment of the paper's contribution.

**Ethical Concerns:**

["NO or VERY MINOR ethics concerns only"]

**Final Justification:**

I have read the comments by the other reviewers and the way the authors have engaged with them.  I have upped my scores accordingly.

**Limitations:**

Yes.

**Paper Formatting Concerns:**

None.

**Quality:**

3

**Strengths And Weaknesses:**

The paper is very well written.  It takes the updates of Graves et al from the perspective of Lienen et al and combines these update rules with a proposed entropic accumulator, following which this is adapted to the hierarchical construction of Zuegner et al.  This facilitates the application to the particle physics context, which makes this paper an interesting contribution.  While the results depict what is naturally available given a probabilistic model -- likelihoods and posterior averages -- it would have helped to see a set of conceptual insights that get opened up as a consequence of these powers.

The Ginkgo trees are generated with an angular distance $d_{ij}=\Delta R^2_{ij}$ between two particles $i$, $j$ in mind.  The comparison shown in C.2 Table 4 is against anti-$k_T$ which uses a different $d_{ij}$ using higher values of $p_T$ to act as a seed to agglomeratively create hierarchies.  This is not a fair comparison.  It would be helpful to see the comparisons with Cambridge-Aachen and the $k_T$ algorithm.  Moreover, from the numbers in Table 3, it is clear that the invariant mass plays a key role in outperforming anti-$k_T$.  Could you provide some explanations as to why this might be the case?  Why do the angular features not guarantee l. 968-9 (ii-iii) is satisfied, whereas an unsupervised method appears to do so by virtue of the agglomerative step?  Does this not invalidate the method as a reliable probe of jets that are typically extracted via anti-$k_T$ at the LHC and later processed via C-A and $k_T$ algorithms for substructure and tagging studies, and for pileup removal?  While the application of the generative framework to the particle physics context is elegant, it is not clear what insights this helps extract that is physically meaningful.

---

> ### Author Rebuttal · Authors · 2025-07-30
>
> We thank the reviewer for their thorough and thoughtful feedback. In the following, we address their concerns and comments.
>
> **Comment:** Could you add comparisons with Cambridge-Aachen and the $k_t$ algorithms?
> **Response:** We follow the reviewer's suggestion and include the two baselines. We reported the anti-$k_t$ algorithm because the CA and $k_t$ often violated constraints (ii) in App. B.6, leading to low valid fractions.
>
> Valid fraction:
>
> | Dataset | CA     | $k_t$   | anti-$k_t$ | TreeGen           |
> |---------|--------|-------|----------|------------------------|
> | QCD-S   | 0.416  | 0.302 | 0.840    | **0.997 $\pm$ 0.001**       |
> | QCD-M   | 0.088  | 0.050 | 0.552    | **0.977 $\pm$ 0.010**       |
> | QCD-L   | 0.025  | 0.008 | 0.440    | **0.943 $\pm$ 0.016**       |
>
> LLH fraction:
>
> | Dataset | CA     | $k_t$   | anti-$k_t$ | TreeGen           |
> |---------|--------|-------|----------|------------------------|
> | QCD-S   | 0.989  | **1.029** | 0.873    | 1.003 $\pm$ 0.002       |
> | QCD-M   | 0.987  | **1.050** | 0.752    | 0.994 $\pm$ 0.002       |
> | QCD-L   | 0.978  | **1.065** | 0.636    | 0.975 $\pm$ 0.001       |
>
>
> As we observe, the CA and $k_t$ algorithms achieve higher LLH fractions than the anti-$k_t$ algorithm but come with a lower validity.
>
> **Comment:** From the numbers in Table 3, it is clear that the invariant mass plays a key role in outperforming anti-$k_t$. Could you provide some explanations as to why this might be the case?
> **Response:** The invariant mass features play a crucial role in the validity of the trees. This results from the conditions (ii) and (iii) in App. B.6, inducing constraints on the invariant mass of the trees.
>
> A single misconnected branch, i.e., split, is sufficient to violate the validity of a tree. Therefore, the invariant mass features become substantial for the valid fraction, while the LLH only receives a minor gain.
>
> **Comment:** Why do the angular features not guarantee L968-9 (ii-iii) is satisfied, whereas an unsupervised method appears to do so by virtue of the agglomerative step?
> **Response:** While the angular features are theoretically sufficient, we noticed that the information is more easily accessible if further processed. Furthermore, while valid branches are essential for the evaluation, our training loss does not prioritize these. For illustration, we have added the final training loss in Tab. 3:
>
> |Model|Valid Frac.|LLH Frac.| Training Loss |
> |--|--|--|--|
> | Base model| 0.563 $\pm$ 0.004| 0.937 $\pm$ 0.002| 0.362 $\pm$ 0.003 |
> | + Ancestor Features| 0.564 $\pm$ 0.001| 0.939 $\pm$ 0.001| 0.361 $\pm$ 0.004 |
> | + 4-Momentum + Angular Features  | 0.645 $\pm$ 0.021| 0.982 $\pm$ 0.001| 0.334 $\pm$ 0.002 |
> | + Invariant Mass | 0.997 $\pm$ 0.001 | 1.003 $\pm$ 0.002 | 0.319 $\pm$ 0.002 |
>
> As we observe, including 4-Momentum and angular features reduces the loss but only slightly improves validity. Including the invariant mass, however, substantially improves the valid fraction with a minor impact on the loss.
>
>
> **Comment:** Does this not invalidate the method as a reliable probe of jets that are typically extracted via anti-$k_t$ at the LHC and later processed via C-A and $k_t$ algorithms for substructure and tagging studies, and for pileup removal?
> **Response:** TreeGen offers access to the full posterior over trees, something deterministic algorithms lack. Even if later steps use CA/$k_t$, TreeGen can provide multiple high‑probability trees and potential uncertainty estimates for downstream analyses.
>
> **Comment:** Please provide potentially physically meaningful insights that this, otherwise elegant, approach might help extract. Given the concerns I have raised, being able to appreciate what this way of processing jets might bring to typical tasks undertaken would enhance my assessment of the paper's contribution.
> **Response:** The key insight of our approach is the feasibility of modelling jet clustering as a generative task. While classical agglomerative jet algorithms output one tree per event, TreeGen models a distribution over all potential hierarchies consistent with the observed four‑momentum. This probabilistic framework provides two key advantages over deterministic trees:
>
> 1. Deterministic approaches only provide a single tree covering only a minor part of the posterior distribution, as many different splitting processes can lead to the same observed leaves. This is especially important for large trees due to the combinatorial explosion.
> 2. Modelling the conditional trees in a distribution provides empirical uncertainty estimates via the empirical sampled distribution. The samples can then be filtered by specific desired properties, e.g., validity or high likelihoods. Our likelihood maximization experiment provides an example (see Fig. 7).
>
> Moreover, TreeGen provides better scalability than exact posterior methods [1], while faithfully approximating the posterior (see Fig. 6). Finally, future work could explore imputation/inpainting tasks inferring only parts of the tree.
>
> We again thank the reviewer for their feedback and are confident that we were able to improve the manuscript.
>
> [1] **Macaluso, Sebastian, Craig Greenberg, Nicholas Monath, Ji Ah Lee, Patrick Flaherty, Kyle Cranmer, Andrew McGregor, and Andrew McCallum.** "Cluster trellis: Data structures & algorithms for exact inference in hierarchical clustering." In International Conference on Artificial Intelligence and Statistics, pp. 2467-2475. PMLR, 2021.

---

> > ### Comment · Reviewer_tb6N · 2025-08-01
> > **Satisfied**
> >
> > Thanks for engaging with and responding to the questions.  I found the responses thorough and addressed my concerns.  I think the paper should be accepted.

---

> > > ### Author Response · Authors · 2025-08-02
> > >
> > > Thank you for your response. We are glad your concerns have been addressed and remain available for any further questions or suggestions.

---

### Official Review · Reviewer_Za3S · 2025-07-02

**Clarity:** 3
**Significance:** 2
**Originality:** 2
**Rating:** 3
**Confidence:** 4

**Summary:**

How should we model distributions over hierarchies? This long addressed question is approached with Bayesian Flow Networks and a novel scheduler for smooth and consistent entropy decay across varying numbers of categories. The authors describe their probabilistic framework, present the application for clustering particles in jet physics. The proposed method uses bayesian updates and gradient-based learning for the feature encoder.

**Questions:**

* Can you describe more why some of the past work such as "Hierarchical clustering in particle physics through reinforcement learning" do not provide ways to sample hierarchies?
* Can you say more about the assumptions of the given distribution over trees? Should other models be considered?
* What happens if no simulator is available? How should we think about OOD data?

**Ethical Concerns:**

["NO or VERY MINOR ethics concerns only"]

**Final Justification:**

The authors address much of my concerns. Given the support of other reviewers and the addressing of comments, I am quite torn about whether or not to raise my score. However, I am not able to champion the paper myself and so I leave my original rating. That said, I am OK if the paper is accepted.

**Limitations:**

yes

**Quality:**

2

**Strengths And Weaknesses:**

TreeGen is an interesting paper, which explores the intersections of probabilistic methods for discovering hierarchical structure, feature learning, and applications in niche and intriguing areas.

Strengths of the paper include:
* Quality: The paper reads in a complete way, experiments and hypotheses are reasonable and explored adequately.
* Clarity: The paper reads clearly, explains this in an understandable way.

Weaknesses include:
* Quality: It is not clear if there should be more comparison & discussion around the assumptions of the distribution over trees given by treegen's model compared to other families of models considered.
* Originality: The differentiating factors seem to be relatively minor (though crucial for performance) (at least as I understand) compared BFNs and past work. It is also not a new application of machine learning. I see this as the key weakness. The methodology seems to be only minor deviations from the past work, the applications are not especially novel either.
* Significance: There are certainly people who are deeply focused on the kinds of problems in this paper, however, it is not clear how significant the results are to a broader audience. The advances on BFNs seem relatively minor

Missing references:
* [Variational Pseudo Marginal Methods for Jet Reconstruction in Particle Physics](https://openreview.net/forum?id=pCapRF2vFf) - seems to be a key reference

---

> ### Author Rebuttal · Authors · 2025-07-30
>
> We thank the reviewer for their valuable feedback. In the following, we address their comments.
>
> **Comment:** Quality: It is not clear if there should be more comparison & discussion around the assumptions of the distribution over trees given by treegen's model compared to other families of models considered.
> **Response:** The only assumption of the distribution is that the structures are directed acyclic graphs, i.e., trees/hierarchies. Both CatFlow and BFN impose the same restrictions. In contrast, the agglomerative baselines are tailored to the jet clustering task and only produce binary hierarchies. We will clarify this in the updated manuscript.
>
>
> **Comment:** Originality: The differentiating factors seem to be relatively minor (though crucial for performance) compared to BFNs and past work. It is also not a new application of machine learning.
> **Response:** We respectfully disagree with the reviewer and would like to reiterate the contributions and novelty of our paper to highlight differences from previous works.
> First, our model results from a simplified theoretical perspective and derivation. Bayesian Flow Networks are derived by introducing an input, output, sender, receiver, flow, and update distribution. TreeGen, however, does not require these and is derived via a simple Bayesian update, yielding a different training loss and update equations.
> Second, we adapt generative models to hierarchies by drawing connections to the clustering literature [9], introducing an inductive bias ensuring valid hierarchies with a novel scheduler balancing the entropy (see Sec. 3).
> Finally, unlike dominant greedy approaches, we present a probabilistic perspective on jet clustering which learns the full posterior distribution over trees and demonstrate that TreeGen faithfully learns the posterior distribution with substantial performance gains over previous works.
>
>
> **Comment:** It is unclear how significant the results are to a broader audience. The advances on BFNs seem relatively minor
> **Response:** Although the resulting model resembles a similar update compared to BFNs, they are derived from a simplified perspective and yield substantial performance improvements, i.e., on the larger W-Boson jets dataset, the valid fraction increases by nearly 60%.
>
> Our results are important to:
> 1. High-energy physics, where TreeGen improves over traditional jet-clustering algorithms.
> 2. Hierarchical modelling, since our method is not limited to jets but can be applied to any hierarchical data.
> 3. Generative modelling, as our derivation simplifies BFNs and allows for a broader range of modelling choices with substantial empirical improvements.
>
>
> **Comment:** Can you describe more why some of the past work such as "Hierarchical clustering in particle physics through reinforcement learning" do not provide ways to sample hierarchies?
> **Response:**
> Most existing methods search for the maximum-likelihood tree $\mathcal{T}^* = \operatorname{arg} \max_{\mathcal{T}} p(\mathcal{T}\mid X)$. Algorithms such as [6,7,8] frame this as a clustering problem and build the tree bottom-up. Often, these algorithms are deterministic and focus on the small subset of high likelihood trees. For this reason, the constructed trees are not samples from the posterior $p(\mathcal{T} \mid X)$.
> Sampling from the posterior $p(\mathcal{T}\mid X)$ is hard because the number of candidate trees grows doubly exponentially with the number of leaves, limiting earlier approaches such as Cluster Trellis [4] to trees with fewer than 20 elements. Making use of generative modelling, TreeGen is able to produce samples from the approximated posterior for trees in the order of 100 elements.
>
> **Comment:** Can you say more about the assumptions of the given distribution over trees? Should other models be considered?
> **Response:**
> Our parametrization is tailored to directed acyclic graphs. We do not place any assumptions or constraints on the trees in the distribution, and they can have varying sizes and branching factors.
>
> **Comment:** What happens if no simulator is available? How should we think about OOD data?
> **Response:** In high-energy physics, there are typically various simulators available [1,2,3]. However, if no simulators are available, one can first gather a small set of samples from the ground-truth posterior if the probability of splits is known [4], serving as a training set. Alternatively, if neither of these is available, one could apply reweighted importance sampling, which only requires an unnormalized energy function [5] (Appendix D.2). In the case of OOD data, TreeGen can be finetuned on real observations, which we leave for future work.
>
>
> We again thank the reviewer for their valuable feedback and have included the missing reference in the updated manuscript. We are happy to address any remaining and upcoming concerns.
>
> [1] **Bellm, Johannes, Stefan Gieseke, David Grellscheid, Simon Plätzer, Michael Rauch, Christian Reuschle, Peter Richardson et al.** "Herwig 7.0/Herwig++ 3.0 release note." The European Physical Journal C 76, no. 4 (2016): 196.
>
> [2] **Bierlich, Christian, Smita Chakraborty, Nishita Desai, Leif Gellersen, Ilkka Helenius, Philip Ilten, Leif Lönnblad et al.** "A comprehensive guide to the physics and usage of PYTHIA 8.3." SciPost Physics Codebases (2022): 008.
>
> [3] **Agostinelli, Sea, John Allison, K. al Amako, John Apostolakis, Henrique Araujo, Pedro Arce, Makoto Asai et al.** "Geant4—a simulation toolkit." Nuclear instruments and methods in physics research section A: Accelerators, Spectrometers, Detectors and Associated Equipment 506, no. 3 (2003): 250-303.
>
> [4] **Macaluso, Sebastian, Craig Greenberg, Nicholas Monath, Ji Ah Lee, Patrick Flaherty, Kyle Cranmer, Andrew McGregor, and Andrew McCallum.** "Cluster trellis: Data structures & algorithms for exact inference in hierarchical clustering." In International Conference on Artificial Intelligence and Statistics, pp. 2467-2475. PMLR, 2021.
>
> [5] **Tong, Alexander, Kilian Fatras, Nikolay Malkin, Guillaume Huguet, Yanlei Zhang, Jarrid Rector-Brooks, Guy Wolf, and Yoshua Bengio.** "Improving and generalizing flow-based generative models with minibatch optimal transport." arXiv preprint arXiv:2302.00482 (2023).
>
> [6] **Greenberg, C.S., Macaluso, S., Monath, N., Dubey, A., Flaherty, P., Zaheer, M., Ahmed, A., Cranmer, K. &amp; McCallum, A.**. (2021). “Exact and approximate hierarchical clustering using A*“.
>
> [7] **Brehmer, Johann et al.** “Hierarchical clustering in particle physics through reinforcement learning.” ArXiv abs/2011.08191 (2020).
>
> [8] **Greenberg, Craig S., Sebastian Macaluso, Nicholas Monath, Ji-Ah Lee, Patrick Flaherty, Kyle Cranmer, Andrew McGregor, and Andrew McCallum**. "Data Structures & Algorithms for Exact Inference in Hierarchical Clustering." arXiv preprint arXiv:2002.11661 (2020).
>
> [9] **Zügner, Daniel, Bertrand Charpentier, Morgane Ayle, Sascha Geringer, and Stephan Günnemann.** "End-to-end learning of probabilistic hierarchies on graphs." In International Conference on Learning Representations. 2021.

---

> > ### Author Response · Authors · 2025-08-06
> >
> > Dear reviewer,
> >
> > We would like to thank you again for reviewing our paper and providing valuable feedback to improve our work.
> >
> > In our rebuttal, we addressed the raised comments and answered the remaining questions. As we approach the conclusion of the discussion period, we would greatly appreciate hearing whether your concerns have been addressed. We remain fully available to provide clarifications or answer any additional questions you may have.

---

### Official Review · Reviewer_yzpo · 2025-07-03

**Clarity:** 2
**Significance:** 2
**Originality:** 1
**Rating:** 4
**Confidence:** 3

**Summary:**

The paper develops a probabilistic generative framework for hierarchies, named TreeGen. TreeGen is built on top of two earlier frameworks: Bayesian Flow Networks (BFNs) and Bayesian Sample Inference (BSI), after adapting for the tree-input data type. Moreover, a logarithmic entropy scheduler is proposed that allows keeping hyperparameters constant across all sizes of hierarchies. Finally, the performance achieved by the new framework is showcased on Jet Clustering experiments, yielding superior performance compared to CatFlow and standard BFN baselines.

**Questions:**

1. The contribution that the paper makes on top of the existing BSI and BFN frameworks seems incremental, and I'm not convinced this validates acceptance as a conference paper. As spelled out in A1, differences from BFN amount to slightly different dynamics over sampling and a different loss over training. It would be helpful that differences from BSI/BFN are clearly articulated over the derivation of TreeGen.
2. Despite including the entropy scheduler as part of the main contributions of the work, I could not find ablation studies that empirically reinforce its usefulness on TreeGen and the considered baselines.
3. The likelihood maximization experiment contains a single unsophisticated baseline. Could authors also include the achieved performance by the probabilistic baselines considered in the rest of the experiments (CatFlow, standard BFNs)?

**Ethical Concerns:**

["NO or VERY MINOR ethics concerns only"]

**Final Justification:**

Thanks to the authors for their copious response and further experimentation. I have now increased my score accordingly.

**Limitations:**

yes

**Quality:**

3

**Strengths And Weaknesses:**

**Strengths**
1. The significance of the paper is timely for the Jet Clustering problems. The experimental results are mostly convincing, showing improvement upon state-of-the-art methods.
2. The paper is well-written in general. A good job is done at motivating the problem and presenting related work.

**Weaknesses**
Please see below.

---

> ### Author Rebuttal · Authors · 2025-07-30
>
> We thank the reviewer for their thorough feedback. We address their remarks in the following.
>
> **Comment:** It would be helpful that differences from BSI/BFN are clearly articulated over the derivation of TreeGen.
> **Response:** We would like to reiterate our contributions and differences from previous works. First, we derive a categorical of BSI, which uses the Bayesian update proposed by BFNs [1], but obtain a different loss and update rule (see App. A.1). Second, we simplify the theoretical framework by omitting the input distribution, output distribution, sender distribution, receiver distribution, flow distribution, and update distribution. Third, we extend this categorical framework to hierarchies by connecting it to the hierarchical clustering literature [2], which has not been done before. Finally, we present a probabilistic view of jet clustering and demonstrate that TreeGen empirically outperforms prior methods by a substantial margin.
>
> We will highlight these similarities and differences in section 3 of the updated manuscript.
>
> **Comment:** The proposed entropy scheduler is not ablated.
> **Response:** We agree with the reviewer that this ablation could be interesting and have conducted additional experiments. However, our entropy scheduler was not designed to outperform the scheduler by Graves et al. [1], but rather to eliminate the need to retune it for different experiments.
>
> In the following, we show experiments conducted with their original scheduler and varying C. In [1], the authors used C=0.75 and C=3.0 in different experiments. Additionally, we tested C=6.0 and C=9.0. The following table shows the results on QCD-S.
>
>
> | Scheduler   | Valid Fraction    | LLH Fraction      |
> |-------------|-------------------|-------------------|
> | Ours        | 0.997 ± 0.001     | 1.003 ± 0.002     |
> | C = 0.75    | 0.410 ± 0.019      | 0.990 ± 0.002      |
> | C = 3.0     | 0.989 ± 0.001     | 1.002 ± 0.002     |
> | C = 6.0     | 0.994 ± 0.001     | 1.003 ± 0.002     |
> | C = 9.0     | 0.992 ± 0.004     | 1.002 ± 0.001 |
>
> The scheduler of Graves et al. [1] is largely influenced by C and reaches matching performance after tuning. We will include these results in the updated manuscript.
>
>
>
> **Comment:** The likelihood maximization experiment contains a single unsophisticated baseline. Could authors also include the achieved performance by the probabilistic baselines (CatFlow, standard BFNs) considered in the rest of the experiments?
> **Response:** We follow the reviewer's suggestion and include the experiments in Figure 7 in the updated manuscript. In the following table, we present the maximum log-likelihoods for the different methods after $N$ samples:
>
>
> | Model | 1 | 5 | 10 | 15 | 20 | 25 | 30 | 35 | 40 | 45 | 50 |
> | -- | --  | --  | --  | --  | --  | --  | --  | --  | --  | --  | --  |
> | CatFlow | -48.267 $\pm$ 0.072 | -47.973 $\pm$ 0.026 | -47.219 $\pm$ 0.085 | -46.879 $\pm$ 0.072 | -46.669 $\pm$ 0.056 | -46.504 $\pm$ 0.061 | -46.405 $\pm$ 0.057 | -46.323 $\pm$ 0.047 | -46.256 $\pm$ 0.044 | -46.194 $\pm$ 0.047 | -46.148 $\pm$ 0.046 |
> | BFN | -49.151 $\pm$ 0.100 | -47.224 $\pm$ 0.022 | -46.597 $\pm$ 0.027 | -46.324 $\pm$ 0.030 | -46.153 $\pm$ 0.025 | -46.038 $\pm$ 0.024 | -45.946 $\pm$ 0.027 | -45.881 $\pm$ 0.026 | -45.822 $\pm$ 0.029 | -45.778 $\pm$ 0.028 | -45.741 $\pm$ 0.028 |
> | TreeGen | **-48.531** $\pm$ 0.084 | **-46.476** $\pm$ 0.054 | **-46.011** $\pm$ 0.067 | **-45.810** $\pm$ 0.059 | **-45.697** $\pm$ 0.045 | **-45.610** $\pm$ 0.046 | **-45.548** $\pm$ 0.045 | **-45.496** $\pm$ 0.041 | **-45.460** $\pm$ 0.038 | **-45.429** $\pm$ 0.036 | **-45.402** $\pm$ 0.036 |
>
>
> Greedy: -45.746
>
> As shown, TreeGen consistently outperforms the two baselines, BFN and CatFlow. Notably, BFN requires 50 samples to match the log-likelihood of the greedy baseline, while CatFlow remains below its performance. This again highlights the advancements of TreeGen over BFNs.
>
>
> We again thank the reviewer for their feedback and are happy to address any remaining concerns.
>
> [1] **Graves, Alex, Rupesh Kumar Srivastava, Timothy Atkinson, and Faustino Gomez.** "Bayesian flow networks." arXiv preprint arXiv:2308.07037 (2023).
>
> [2] **Zügner, Daniel, Bertrand Charpentier, Morgane Ayle, Sascha Geringer, and Stephan Günnemann.** "End-to-end learning of probabilistic hierarchies on graphs." In International Conference on Learning Representations. 2021.

---

### Decision · Program_Chairs · 2025-09-17

**Decision:**

Accept (poster)

**Comment:**

This paper describes a generalization of Bayesian Flow Networks to model hierarchical data, as well as a new entropy scheduling strategy, and experimental results for jet clustering problems. Informally speaking, the method starts with a probabilistic hierarchy which is gradually reduced in entropy until a discrete hierarchy is found. Reviewers agreed the method was novel, the paper was clearly written, and the experimental results were of high quality, plausibly demonstrating a practical advance.